# Structural determinants of voltage-gating properties in calcium channels

**Monica L Fernández-Quintero[1,2†], Yousra El Ghaleb[1†], Petronel Tuluc[3], Marta Campiglio[1], Klaus R Liedl[2], Bernhard E Flucher[1]\***

[1]Department of Physiology and Medical Physics, Medical University Innsbruck, Innsbruck, Austria; [2]Department of General, Inorganic and Theoretical Chemistry, University of Innsbruck, Innsbruck, Austria; [3]Department of Pharmacology and Toxicology, Institute of Pharmacy and Center for Molecular Biosciences, University of Innsbruck, Innsbruck, Austria

**Abstract** Voltage-gated calcium channels control key functions of excitable cells, like synaptic transmission in neurons and the contraction of heart and skeletal muscles. To accomplish such diverse functions, different calcium channels activate at different voltages and with distinct kinetics. To identify the molecular mechanisms governing specific voltage sensing properties, we combined structure modeling, mutagenesis, and electrophysiology to analyze the structures, free energy, and transition kinetics of the activated and resting states of two functionally distinct voltage sensing domains (VSDs) of the eukaryotic calcium channel $Ca_V1.1$. Both VSDs displayed the typical features of the sliding helix model; however, they greatly differed in ion-pair formation of the outer gating charges. Specifically, stabilization of the activated state enhanced the voltage dependence of activation, while stabilization of resting states slowed the kinetics. This mechanism provides a mechanistic model explaining how specific ion-pair formation in separate VSDs can realize the characteristic gating properties of voltage-gated cation channels.

**\*For correspondence:** bernhard.e.flucher@i-med.ac.at

[†]These authors contributed equally to this work

**Competing interests:** The authors declare that no competing interests exist.

## Introduction

Voltage-gated calcium channels ($Ca_V$) translate membrane depolarization into calcium influx. Thus, they contribute to cellular excitability and they couple electrical activity to fundamental cell functions like contraction of heart and skeletal muscle, secretion of neurotransmitters and hormones, and the regulation of gene expression. Together with voltage-gated sodium channels ($Na_V$), $Ca_V$s form a structurally related ion channel superfamily with a fourfold symmetry (*Figure 1A*). Their pore-forming $\alpha_1$ subunits are composed of four homologous but non-identical domains (repeats I-IV), each containing six transmembrane helices (S1-S6). The S5 and S6 helices plus the connecting P loop of all four repeats form the central channel pore with the selectivity filter and the activation gate (*Catterall et al., 2020*). Helices S1-S4 of each repeat form separate voltage sensing domains (VSDs). The S4 helix contains positively charged residues (termed gating charges) in every third position, and its movement across the electric field upon membrane depolarization is thought to initiate the conformational change resulting in channel opening (*Catterall et al., 2017*).

Several high-resolution structures of prokaryotic and eukaryotic $Na_V$ channels have been solved (*Lenaeus et al., 2017*; *Pan et al., 2018*; *Payandeh et al., 2011*; *Yan et al., 2017*; *Zhang et al., 2012*). Recent advances in cryo-electron microscopy (cryo-EM) enabled the determination of the structure of the voltage-gated calcium channel $Ca_V1.1$ at 3.6 Å resolution, displaying a closed pore and the VSDs in the activated up-state (*Wu et al., 2016*; *Wu et al., 2015*). Very recently, three cryo-EM structures of homo-tetrameric sodium channels experimentally locked in resting (or VSD-down) states have been reported (*Wisedchaisri et al., 2021*; *Wisedchaisri et al., 2019*; *Xu et al., 2019*). However, up to now the resting states of eukaryotic $Ca_V$ and $Na_V$ channels remained inaccessible to

 

**eLife digest** Communication in our body runs on electricity. Between the exterior and interior of every living cell, there is a difference in electrical charge, or voltage. Rapid changes in this so-called membrane potential activate vital biological processes, ranging from muscle contraction to communication between nerve cells.

Ion channels are cellular structures that maintain membrane potential and help 'excitable' cells like nerve and muscle cells produce electrical impulses. They are specialized proteins that form highly specific conduction pores in the cell surface. When open, these channels let charged particles (such as calcium ions) through, rapidly altering the electrical potential between the inside and outside the cell.

To ensure proper control over this process, most ion channels open in response to specific stimuli, which is known as 'gating'. For example, voltage-gated calcium channels contain charge-sensing domains that change shape and allow the channel to open once the membrane potential reaches a certain threshold. These channels play important roles in many tissues and, when mutated, can cause severe brain or muscle disease.

Although the basic principle of voltage gating is well-known, the properties of individual voltage-gated calcium channels still vary. Different family members open at different voltage levels and at different speeds. Such fine-tuning is thought to be key to their diverse roles in various parts of the body, but the underlying mechanisms are still poorly understood. Here, Fernández-Quintero, El Ghaleb et al. set out to determine how this variation is achieved.

The first step was to create a dynamic computer simulation showing the detailed structure of a mammalian voltage-gated calcium channel, called $Ca_V1.1$. The simulation was then used to predict the movements of the voltage sensing regions while the channel opened.

The computer modelling experiments showed that although the voltage sensors looked superficially similar, they acted differently. The first of the four voltage sensors of the studied calcium channel controlled opening speed. This was driven by shifts in its configuration that caused oppositely charged parts of the protein to sequentially form and break molecular bonds; a process that takes time. In contrast, the fourth sensor, which set the voltage threshold at which the channel opened, did not form these sequential bonds and accordingly reacted fast. Experimental tests in muscle cells that had been engineered to produce channels with mutations in the sensors, confirmed these results.

These findings shed new light on the molecular mechanisms that shape the activity of voltage-gated calcium channels. This knowledge will help us understand better how ion channels work, both in healthy tissue and in human disease.

experimental structure determination. Nevertheless, many years of experimental work and structure modeling provide ample support for the sliding helix model of the voltage sensor action (*Catterall et al., 2017*; *Yarov-Yarovoy et al., 2012*). According to this model, the negative membrane potential at rest pulls the positively charged S4 helices down toward the cytoplasmic side of the membrane holding the channel gate closed. The reversal of the electric field upon membrane depolarization causes the outward displacement of the S4 helix by about 10 Å. The movement of two to three positive gating charges through the hydrophobic constriction site (HCS) in the center of the VSD is facilitated by the transient formation of ion-pair interactions with negative countercharges in the other helices of the VSD (*Catterall et al., 2017*).

While this model describes the principal mode of voltage sensor action, without further structure-function data, it does not explain how the four homologous but structurally distinct VSDs of eukaryotic channels cooperate in channel gating and how the unique gating properties of different channel isoforms are achieved. The distinct structure of the four VSDs of eukaryotic $Na_V$ and $Ca_V$ suggests that there might be considerable variability between the four VSDs of a channel in the movement of the S4 helices and the molecular interactions of their gating charges. In fact, accumulating evidence indicates functional differences between the VSDs of individual channel isoforms (*Ahern et al., 2016*; *Pantazis et al., 2014*; *Tuluc et al., 2016a*). The rabbit skeletal muscle $Ca_V1.1$ is the first member of the $Ca_V$ family for which the structure has been solved (*Wu et al., 2016*; *Wu et al., 2015*;

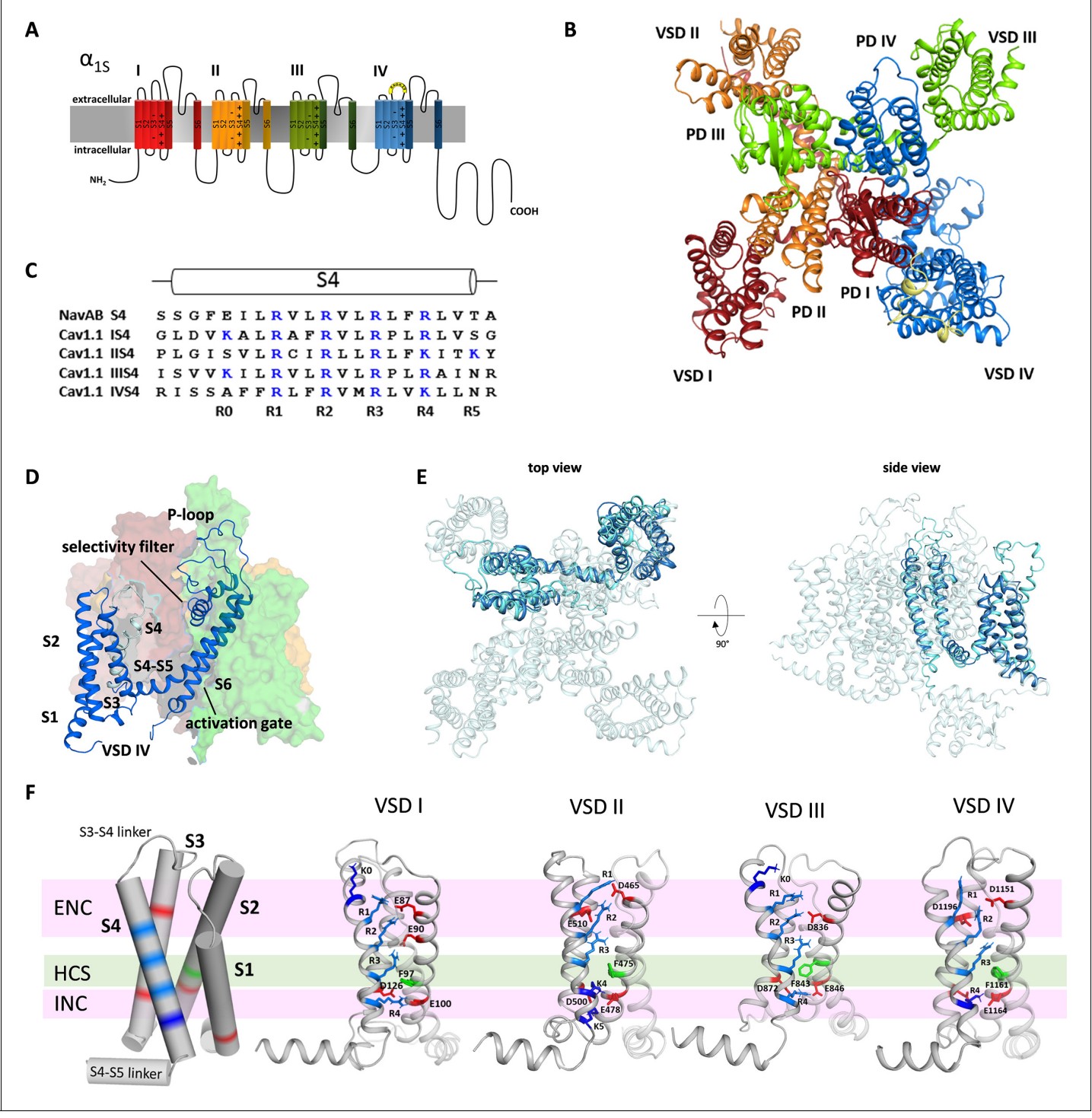

**Figure 1.** Structure model of the hetero-tetrameric human Ca$_V$1.1. (**A**) Domain structure of eukaryotic Ca$_V$ channels. (**B**) Structure model of the human Ca$_V$1.1 $\alpha_1$ subunit (top view; color code as in A) refined with molecular dynamics (MD) simulation in a membrane environment (see 'Materials and methods') based on the 3.6 Å structure of rabbit Ca$_V$1.1 (*Pan et al., 2018*; *Wu et al., 2016*). (**C**) Sequence alignment of the S4 helices of each Ca$_V$1.1 voltage sensing domain (VSD) compared to the homo-tetrameric Na$_V$Ab; gating charges (R, K) are indicated in blue. (**D**) Structure of a single repeat (IV) within the space-filling model of Ca$_V$1.1. (**E**) Structural overlay of Na$_V$Ab with VSD IV of Ca$_V$1.1. (**F**) Cylindrical representation of the VSD structure showing the positive gating charges in S4 (blue) and countercharges (red) of the intra- and extracellular negative clusters (INC, ENC) in S1, S2, and S3. The phenylalanine in S2, marking the hydrophobic constriction site (HCS), is indicated in green. Ribbon models of the four VSDs of Ca$_V$1.1 in the up-state, showing the side chains of the S4 gating charges (R, light blue; K, dark blue) and their putative ion-pair partners (red). Note that the numbers and positions of the ion-pair interactions in the ENC differ between the VSDs.

*Figure 1 continued on next page*

*Figure 1 continued*

The online version of this article includes the following figure supplement(s) for figure 1:

**Figure supplement 1.** Structure comparison of the four voltage sensing domains (VSDs) of Ca$_V$1.1 with the VSD of Na$_V$Ab.

**Figure supplement 2.** Structure comparison between the four voltage sensing domains (VSDs) of Ca$_V$1.1.

*Zhao et al., 2019*). Biophysically it is characterized by slow kinetics and right-shifted voltage dependence of activation (*Tuluc et al., 2016a*). Together these attributes make Ca$_V$1.1 a prime candidate for studying how specific structural features of the VSDs determine voltage dependence and kinetics of channel activation.

Here, we applied molecular dynamics (MD) simulation and Markov state modeling (MSM) combined with site-directed mutagenesis and electrophysiological analyses to identify the molecular mechanism by which individual VSDs determine the characteristic voltage dependence and kinetics of current activation. Our structure models of the activated and resting states of Ca$_V$1.1 VSDs I and IV are consistent with the sliding helix model and yielded reliable predictions of the importance of ion-pair formation between the outer gating charges and various countercharges within the particular VSD. Our data provide novel insight in how the stabilization of the VSDs in resting and/or the activated states shapes the kinetics and voltage dependence of activation, respectively.

## Results

### The structure of Ca$_V$1.1 reveals differences between VSDs

Based on the cryo-EM structure of Ca$_V$1.1 (*Wu et al., 2016*; *Wu et al., 2015*), we generated a new structural model to study the molecular mechanisms determining the specific gating properties of this voltage-gated calcium channel. To this end, we used the *Rosetta* computational modeling software (*Bender et al., 2016*; *Rohl et al., 2004*) to build a homology model of the human eukaryotic Ca$_V$1.1 and included all missing loops and modeled both splice variants with and without exon 29 (*Tuluc et al., 2009*). The resulting models were equilibrated and simulated at 300 K in the membrane environment to identify favorable side-chain orientations and to relax the protein.

Ca$_V$1.1 is a pseudo-tetrameric channel with a domain-swapped arrangement in which each VSD (S1-S4) is positioned next to the pore domain (S5-S6) of the adjacent repeat in a clockwise orientation (*Catterall et al., 2017*, *Figure 1B,D*). The structures of the individual repeats closely resemble the crystal structure of Na$_V$Ab (C$\alpha$ root-mean-square deviation [RMSD] of 2.7 Å) (*Payandeh et al., 2011*), which is regarded as phylogenetic ancestor of Ca$_V$ and Na$_V$ channels and for which considerable structural information is available (*Figure 1E*; *Figure 1—figure supplement 1*). While all four VSDs of Ca$_V$1.1 display the canonical voltage sensor fold, individually they differ from one another in significant aspects like the length of the helical structures and the number of gating charges in S4 (*Figure 1—figure supplement 2*). Only VSD IV contains four gating charges (R1-R4) at the three-residue interval, like Na$_V$Ab (*Figure 1C*). VSDs I and III possess an additional positive charge (K0) at the outer end of S4, and VSD II an additional gating charge (K5) at the cytoplasmic side of S4. All four VSDs are in the activated (S4-up) state in that (K0) R1, R2, and R3 are positioned above the phenylalanine (in S2) of the HCS and R4 (and K5) below (*Figure 1F*).

As predicted by the sliding helix model (*Catterall et al., 2017*), the gating charges form ion pairs with countercharges of the extracellular negative cluster (ENC) and intracellular negative cluster (INC). The interactions of the inner gating charges with countercharges of the INC are identical in the four VSDs of Ca$_V$1.1, representing the typical arrangement of the highly conserved charge transfer center (*Tao et al., 2010*). However, between the VSDs, the outer ion-pair interactions differ. Overall, the four VSDs of Ca$_V$1.1 can be grouped into two classes – VSDs I and III, and VSDs II and IV, respectively – each with the same number and position of countercharges in the ENC. While in VSDs I and III, the gating charges form ion pairs with two glutamate residues in the S2 helix, VSDs II and IV gating charges interact with one negative countercharge each in the S2 and S3 helices. Also, in VSDs I and III, the additional outermost gating charges are lysines (K0), whereas in VSDs II and IV, the innermost gating charge is a lysine residue (K4) instead of an arginine (R4) in VSDs I and III. Note that lysine forms only a single interaction with a negative countercharge in the INC as opposed to two formed by arginine. Together these differences in the ion pairs formed by

the gating charges indicate that the activated state of VSDs I and III is considerably more stabilized than that of VSDs II and IV.

## MD simulation and MSM of VSD I in activated and resting states

The interactions between gating charges and their ion-pair partners observed in the structure model based on available cryo-EM structures of Ca$_V$1.1 merely represent a snapshot depicting the endpoint of the voltage sensing process. However, because high-resolution structures of resting states of Ca$_V$ channels are lacking, the molecular details of the steps leading up to VSD activation are still elusive. Exploiting the potential of structure modeling to fill this gap (*Jensen et al., 2012*; *Yarov-Yarovoy et al., 2012*), we applied MD simulation and MSM of individual VSDs to predict the structures, kinetics, and energy levels of resting states. To overcome the high energy barriers and the timescale limitations of MD simulations in the absence of the membrane potential, we used *Umbrella* sampling. This enhanced sampling technique explores the conformational transitions of a VSD as the positively charged S4 helix moves along the likely pathway toward the cytoplasmic side of the VSD, and thus create the seeding points for subsequent MD simulations. For this purpose, the obtained structures were clustered based on a geometrical RMSD criterion resulting in about 50 cluster representatives. These were simulated for 100 ns each (aggregated simulation time close to 5 µs) to obtain unbiased trajectories, which were then projected in a time-lagged independent component analysis (tICA), representing the slowest reaction coordinates. Finally, the kinetic coordinate system provided by the tICA allows calculation of thermodynamics and kinetics by an MSM (*Figure 2—figure supplement 1*).

Using this approach, we modeled the resting state structures of Ca$_V$1.1 VSDs I and IV, because they represent the two structurally distinguishable classes of VSDs (see *Figure 1F*; *Figure 1—figure supplement 2*), and because they differentially regulate the specific gating properties of skeletal muscle calcium currents. VSD I determines Ca$_V$1.1's slow activation kinetics and VSD IV its voltage-dependence of activation (*Nakai et al., 1994*; *Tuluc et al., 2016a*). The latter is regulated by alternative splicing in that exclusion of the 19 amino acids of exon 29 from the extracellular loop linking IVS3 and IVS4 causes a 30 mV left-shift of the voltage dependence of activation and a several-fold increase in current density (*Tuluc et al., 2009*). Previously we demonstrated that the loss of voltage sensitivity upon insertion of exon 29 is caused by the relative lateral displacement of S3 and S4 and the resulting loss of ionic interactions of the outer gating charges (R1, R2) with a single countercharge (D1196) in the S3 helix (*Tuluc et al., 2016b*). Considering the importance of these ionic interactions in VSD IV for regulating the channel's voltage dependence, we hypothesized that similarly the specific kinetic properties of Ca$_V$1.1 may be encoded in the structure, ionic interactions, and the molecular kinetics of the state transitions of VSD I.

The free energy maps calculated for the MD simulation of VSD I, VSD IVe, and VSD IVa each comprised four energy minima (*Figure 2A,E,I*) and the corresponding structures resembled the activated and three resting states, as predicted by the sliding helix model. Across these states, the S4 helix of VSD I described a stepwise downward movement of 15.3 Å, corresponding to three helical turns (*Figure 2B,C*). In the activated state, the gating charges K0, R1, R2, and R3 were above phenylalanine (F97) of the HCS, and only R4 was below it. In the deepest resting state 1, only K0 and R1 were positioned above the HCS, while R2, R3, and R4 were located below it. In all four states, IS4 adopted a shifting stretch of $3_{10}$ helical conformation (*Figure 2—figure supplement 2*), so that the side chains of the gating charges all pointed toward the center of the VSD. As IS4 moved from the activated state to resting state 1, R4, R3, R2, and R1 sequentially formed ion pairs with countercharges of the INC (E100 in IS2 and D126 in IS3), which are part of the highly conserved charge transfer center of voltage-gated cation channels (*Figure 2B*; *Figure 3A*, *Tao et al., 2010*). In all states also the gating charges above and below the HCS formed extensive ionic bonds with negative countercharges in IS1 and IS2. In the resting states 2 and 3 and in the activated state R1, R2, and R3 formed ion bonds with E87 and E90, plus in resting state 3 and the activated state K0 formed an additional ion bond with E76 in the IS1-S2 loop. In resting states 1 and 2, gating charges R3 and R4 formed ion bonds with E49 and E54 at the cytoplasmic end of IS1. This multitude of ionic interactions stabilizes each of the consecutive states and thus strictly delineates the path of IS4 through VSD I upon channel activation and deactivation. Note that in addition to the indicated ionic bonds, the gating charges form transient hydrogen bonds and hydrophobic and polar interactions with several other

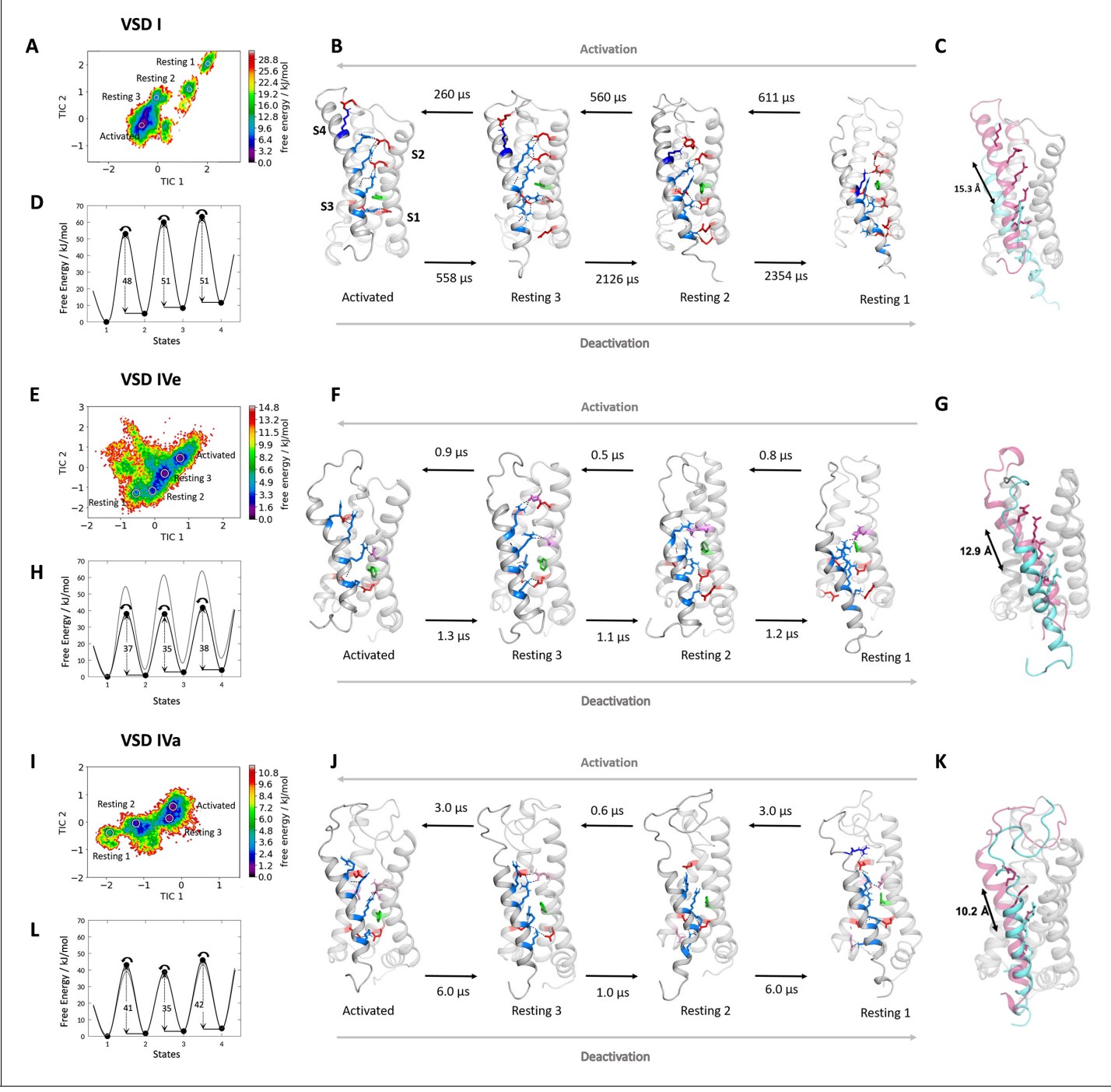

**Figure 2.** Molecular dynamics (MD) simulation and kinetics of voltage sensor transitions of voltage sensing domains (VSDs) I and IV with and without exon 29. (**A,E,I**) The free energy surfaces of 5.0 μs trajectories of VSD I (**A**), VSD IV of Ca$_V$1.1e (**E**), and Ca$_V$1.1a including exon 29 (**I**) reconstructed in the time-lagged independent component analysis (tICA) coordinate space resulted in four macrostates. (**B,F,J**) Representative structures of each VSD in the four macrostates correspond to three resting and the activated states. The S4 gating charges (blue) show a sequential movement relative to the phenylalanine (green) in the hydrophobic constriction site (HCS) and stabilizing interactions with ion-pair partners (red) and H-bond donors/acceptors (pink) in the intracellular negative cluster (INC) and extracellular negative cluster (ENC). Transition kinetics (in μs) were calculated using a Markov state model. (**C,G,K**) Overlays of the activated (magenta) and resting state 1 (cyan) illustrating the maximum displacement of S4 during activation. (**D,H,L**) Schematic 1D representations of the free energy surface of VSD I (**D**), VSD IV of Ca$_V$1.1e (**H**), and Ca$_V$1.1a (**L**), with energy barriers calculated using transition state theory at 0 mV favoring the activated state. Gray trace in (**H**) shows free energy surface of (**D**) for comparison; gray trace in (**L**) shows free energy surface of (**H**) for comparison. Because in skeletal muscle cells, VSDs II and III probably control excitation-contraction coupling (*Flucher, 2020*; *Flucher, 2016*), and as their contribution to channel gating is less well understood, we did not include them in the present study.

*Figure 2 continued on next page*

*Figure 2 continued*

The online version of this article includes the following figure supplement(s) for figure 2:

**Figure supplement 1.** Schematic illustration of our robust protocol for describing and characterizing protein dynamics and statistically validating the results by constructing a Markov state model (MSM).

**Figure supplement 2.** $3_{10}$-Helix content of the S4 helices of Ca$_V$1.1 voltage sensing domains (VSDs) I and IV in the activated and resting state 1.

**Figure supplement 3.** Parameter selection and validation of the Markov state models (MSMs) corresponding to the results presented in *Figures 2*, *4,* and *5*.

**Figure supplement 4.** Structural comparison of the Ca$_V$1.1 voltage sensing domain (VSD) I resting states with the experimentally determined resting state structure of a Na$_V$1.7/Na$_V$Ab chimera.

**Figure supplement 5.** Structural comparison of the Ca$_V$1.1 voltage sensing domain (VSD) IV resting states with the experimentally determined resting state structure of Na$_V$Ab.

putative interaction partners, all of which might contribute to the movement of S4 across the membrane electrical field, but are not subject of the present study.

Next, we used MSM of the MD simulation data to estimate transition times between the resting and activated states during the activating and deactivating VSD motion (*Figure 2B,D*; *Figure 2—figure supplement 1*; *Supplementary file 1*). The conformational transitions between the different activation states of Ca$_V$1.1 VSD I occurred in the high μs to low ms timescale. Because the values calculated in our model are obtained in the absence of the force provided by changes in the electric field, the absolute transition times derived from MSM may not correspond to the actual transition times of the VSD upon physiological activation and deactivation. Nevertheless, relative differences between transition times provide meaningful information when compared between different VSDs or functionally different mutants (see below). Relying on a simple transition state theory model (*Laidler and King, 1983*), we generated a schematic 1D representation of the high-dimensional free energy surface (*Figure 2D*), allowing an intuitive interpretation of free energy levels of the states and ΔG of the energy barriers. The free energy of the activated state was the lowest and transitions in the activating direction were two to four times faster than in the deactivating direction, consistent with the fact that our MD simulations were performed on structure models at a depolarized membrane (0 mV), which favors the activated state of the VSDs. The energy barriers (ΔG) for the three state transitions of VSD I in the activating direction were between 48 and 51 kJ/mol.

## MD simulation and MSM of VSD IV in activated and resting states

How do the molecular interactions during VSD activation and deactivation and the kinetics of state transition differ between VSD I and VSD IV to explain their distinct functions in determining kinetics and voltage dependence of activation, respectively? (*Tuluc et al., 2009*) The basic structural features of the activated and resting states of VSD IV corresponded to those of VSD I, except that in VSD IV the translocation of S4 across the HCS covered a shorter distance and involved fewer ion-pair interactions (*Figure 2F,G,J,K*). Upon the deactivating motion of the S4 helix of VSD IV, only a single gating charge (R3) fully translocated from a position above the HCS (F1161) to below it. Accordingly, the total vertical displacement in the two splice variants of IVS4 was 12.9 Å in VSD IVe and only 10.2 Å in VSD IVa, corresponding to roughly two helical turns.

In the four states of both VSD IV variants, only the two inner gating charges R3 and K4 sequentially interacted with the conserved ion-pair partners of the charge transfer center (E1164 and D1186) (*Figure 2F,J*; *Figure 3A*). R2 moved into the HCS but did not form ion pairs with E1164 or D1186. In the deep resting states, an additional ion pair was formed between K4 and E1121 at the cytoplasmic end of IVS1. At variance with VSD I, ion-pair formation of the outer gating charges of VSD IV was completely absent in the resting states. Instead, R1 and R2 established several weaker hydrogen bonds with side chains of uncharged polar amino acids. In VSD IV ion-pair interactions of R1 and R2 with the ENC were limited to the activated state of the Ca$_V$1.1e splice variant. Within VSD IV they were established with a single ion-pair partner (D1196), the functional importance of which for splicing-dependent regulation of the voltage dependence of activation had been shown previously (*Tuluc et al., 2016b*). Consistent with that study, these interactions were greatly attenuated when exon 29 was included in the IVS3-S4 linker of Ca$_V$1.1a (*Figure 3A,C*). The activated state

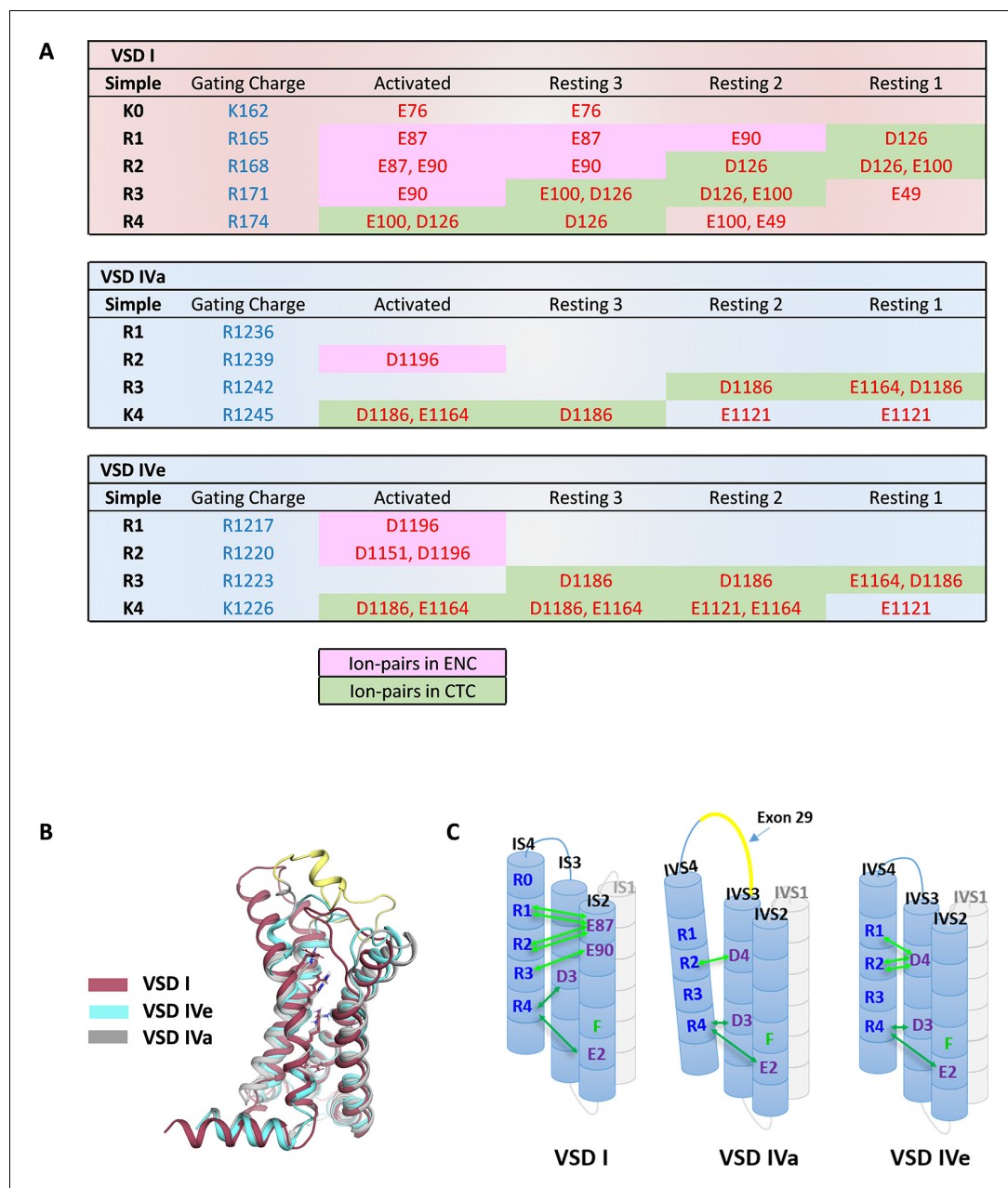

**Figure 3.** Ion-pair partners of Ca$_V$1.1 voltage sensing domains (VSDs) I and IV gating charges in activated and resting states. (**A**) Tabular overview of ion-pair interactions of the positive gating charges (blue) with countercharges (red) observed in the activated and three resting states of VSDs I, IVa, and IVe. Fields shaded in green show the sequential interaction with the ion-pair partners of the intracellular negative cluster (INC). In VSD I this transition through the charge transfer center (CTC) involves R4, R3, R2, R1, whereas in VSD IV only R4 and R3 participate in equivalent interactions. Ion-pair formation with the extracellular negative cluster (ENC) (pink shading) of VSD I involved three gating charges (R3, R2, R1) in the activated and intermediate resting states 2 and 3. In VSD IV ion-pair formation with the ENC is limited to the activated state, and it is further reduced by inclusion of exon 29 in VSD IVa. (**B**) Structure overlay of VSD I, VSD IVa (including exon 29; yellow), and VSD IVe in the activated state. (**C**) Schematic representation of the three VSDs indicating similar ion-pair formation in the INC, representing the conserved CTC (dark green), but highly distinct ion-pair formation in the ENC (light green) of the three analyzed VSDs.

structure of VSD IVa (containing exon 29) showed only a single H-bond between R2 and D1196, compared to two each of R1 and R2 with D1196 in VSD IVe (lacking exon 29). Thus, the three resting states of VSD IV are not stabilized by ion pairs formed by the outer gating charges, and the ion-pairs forming in the activated state are further reduced by inclusion of exon 29 in the linker separating the two participating helices.

Can these striking structural differences explain the kinetic differences conferred to the channel by VSDs I and IV? Our MSM calculations of state transition kinetics strongly support this notion. Compared to VSD I, the energy barriers between the states were substantially lower in VSD IV (35– 38 kJ/mol for IVe and 35–42 kJ/mol for IVa) (*Figure 2H,L*). Consistent with the differences in activation kinetics, the transition times determined for VSD IV were in the µs range (*Figure 2F,J*), which is two to three orders of magnitude faster than those of VSD I. Also, the transitions between resting states 1 and 2, and between resting state 3 and the activated state were about three times faster in $Ca_V1.1a$ compared to $Ca_V1.1e$, indicating an effect of exon 29 insertion on the kinetics of VSD IV movement. However, compared to the substantial difference between the kinetically distinct VSDs I and IV, the differences in energy barriers and transition times between the two VSD IV splice variants remain unexpectedly small. Apparently, the calculated transition times primarily reflect differences in kinetics, but much less differences of the voltage dependence of activation. This makes sense, considering that the height of energy barriers and the transition kinetics are expected to affect the sequential transitions of a VSD through all four states, which determine the channel activation kinetics, whereas changes in voltage sensitivity primarily rely on the stabilization of the activated state and therefore are little affected by differences of the state transitions.

The direct comparison of VSD I and the two splice variants of VSD IV in the activated and resting states demonstrates striking differences between VSDs I and IV in the extent of ion-pair formation in the ENC (*Figure 3*). As shown above, in VSD IV these involve interaction of R1 and R2 with the ion-pair partner D1196 in IVS3 formed in the activated state that is subject to modulation by alternative splicing of exon 29. VSD I lacks an analogous ion-pair partner in the corresponding position of IS3. Instead IS4 displays extensive ion pairs with countercharges (E76, E87, E90) in IS2 that are sequentially formed by gating charges R3, R2, R1, and K0 in the activated and the intermediate resting states 2 and 3 (*Video 1*). This indicates that in VSDs I and IV, the gating charges above the HCS utilize structurally distinct ion-pair partners to stabilize the voltage sensor either only in the activated state (VSD IV) or in the activated and resting states (VSD I). The additionally formed ion pairs in the resting states of VSD I are paralleled by a remarkable increase in the energy barriers and the state transition times, suggesting that the number and strength of interactions between the gating charges and the ENC transiently formed in the resting states determine the slow activation kinetics of VSD I.

## VSD I ion pairs differentially regulate gating properties

To experimentally test this hypothesis, we simultaneously mutated both countercharges, E87 and E90, to alanine (E87A/E90A) in the rabbit GFP-$Ca_V1.1e$ (*Tanabe et al., 1988*; *Tuluc et al., 2009*), expressed them in their native environment in dysgenic myotubes and examined the effects on the gating properties of its calcium currents (*Figure 4A*). The structure of the VSDs in general and in particular the studied residues are highly conserved in $Ca_V$ channels (*Wu et al., 2016*). Immunofluorescence labeling demonstrated that wildtype (WT) and mutant channels were equally expressed and targeted to triad junctions in the myotubes (*Figure 4—figure supplement 1*). In contrast, their gating properties

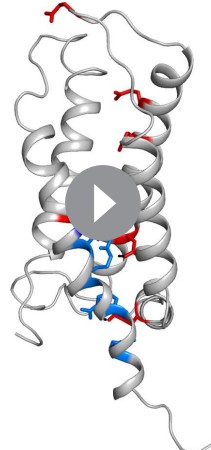

**Video 1.** Movement of wildtype $Ca_V1.1e$ voltage sensing domain (VSD) I upon activation and deactivation, highlighting the ion-pair interactions formed between the S4 gating charges (blue) and relevant countercharges (red) in the S2 and S3 helices.
https://elifesciences.org/articles/64087#video1

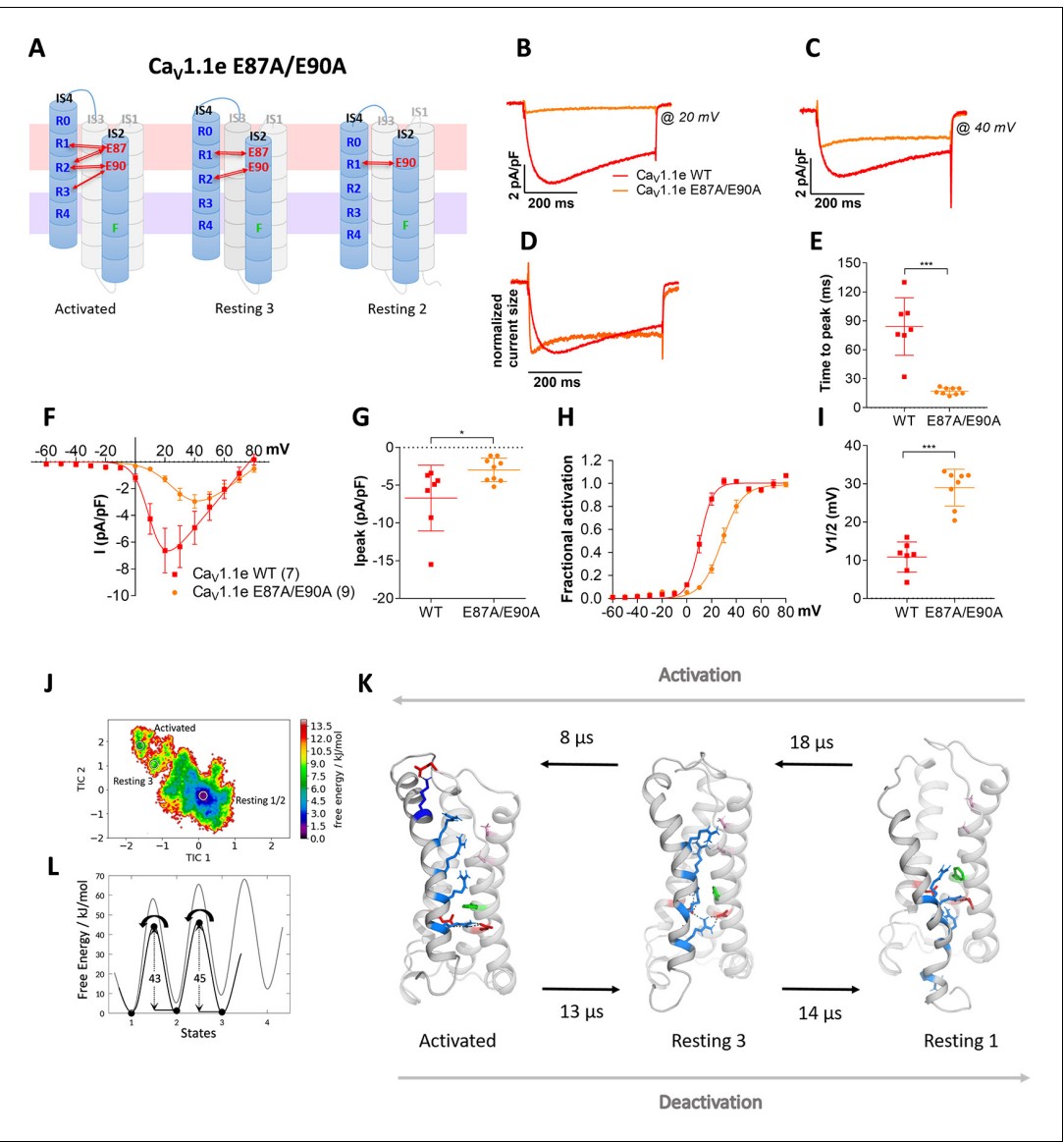

**Figure 4.** Countercharges E87 and/or E90 in IS2 determine the voltage dependence and kinetics of Ca$_V$1.1 current activation and voltage sensing domain (VSD) transitions. (**A**) Schematic model of VSD I in the activated and resting states, showing the putative loss of interactions between gating charges and countercharges E87 and E90 upon their mutation to alanine. (**B–I**) In Ca$_V$1.1e the double mutation E87A/E90A (orange) accelerated activation kinetics and right-shifted the voltage dependence of activation, compared to wildtype (WT) Ca$_V$1.1e (red). (**B,C**) Representative current traces at V$_{max}$ of WT Ca$_V$1.1e (20 mV) and Ca$_V$1.1e E87A/E90A (40 mV), respectively, and normalized currents at V$_{max}$ (**D**). (**E**) Scatter plot of the time to peak; (**F**) current-voltage relationship; (**G**) scatterplot of maximum current density (p=0.03); (**H**) voltage dependence of activation; (**I**) scatter plot of the voltage at half-maximal activation (V½). Mean ± SEM; p-values calculated with Student's t-test, ***p<0.00001. (**J,K**) The time-lagged independent component analysis (tICA) free energy surface of Ca$_V$1.1e E87A/E90A displays three macrostates with structures corresponding to the activated state and resting states 1 and 3, and transition kinetics in the low μs timescale. (**L**) The 1D energy plot shows substantially lower calculated energy barriers between the states of the double mutant (black) compared to the WT VSD I (gray).

The online version of this article includes the following figure supplement(s) for figure 4:

**Figure supplement 1.** Expression and triad targeting of wildtype (WT) and E87A, E90A mutant GFP-Ca$_V$1.1e channels in dysgenic myotubes.

**Figure supplement 2.** Putative countercharge E90, but not E87, determines slow activation kinetics of voltage sensing domain (VSD) I of Ca$_V$1.1e.

differed significantly (*Figure 4B–I*; *Supplementary file 1*). As hypothesized, activation kinetics was more than four times faster in the mutant compared to WT (*Figure 4D,E*), thus identifying E87 and/or E90 as critical determinants of the slow activation kinetics of Ca$_V$1.1. Interestingly, also the voltage dependence of activation was right-shifted to more depolarizing potentials by 18.2 mV and the peak current density was somewhat reduced (*Figure 4F–I*).

If the comparably long transition times for WT VSD I determined by MSM related to the experimentally determined activation kinetics, then MSM of the E87A/E90A mutant channel should result in rapid transition times. This was indeed the case! The transition times of VSD I on activation and deactivation of the E87A/E90A mutant were more than 50 times faster than those of the WT VSD (*Figure 4J–L*; *Figure 2—figure supplement 3*, *Figure 4—figure supplement 2*, *Supplementary file 1*; *Videos 1* and *2*). This supports our interpretation of the role of the two countercharges in determining the gating properties, and also substantiates the reliability and predictive value of the kinetic analysis of our MD simulations. Yet, it is worth noting that the transition kinetics derived from our MD simulation relate to the activation of an isolated VSD, whereas kinetics and voltage dependence of channel activation reflect the concerted action of all four VSDs and its mechanical transduction to the channel gate. Consequently, changes in activation properties of a single VSD will only result in similar changes of current activation, when this VSD is obligatory and rate-limiting for gating, or, in an allosteric model, according to its relative contribution to the gating process. This limitation may also account for the different magnitudes of the effects (50-fold vs. 5-fold) on the activation kinetics of the E87A/E90A mutant observed in MSM and current recordings.

As E87 and/or E90 govern the kinetics as well as voltage dependence of Ca$_V$1.1 activation, we wondered whether these two properties are mechanistically linked to each other or separable? Our structural model predicts that E87 interacts with R1 in resting state 3, and with R1 and R2 in the activated state (*Figure 5A*), which is consistent with a prime role in stabilizing the activated state. In contrast, E90 forms consecutive interactions with R3, R2, and R1 in resting state 2, resting state 3, and the activated state, respectively (*Figure 5K*), thus stabilizing VSD I both in its resting and activated states. To examine the individual contributions of E87 and E90 to shaping the gating properties, we generated constructs with individual E87A and E90A substitutions. The two mutations showed differential effects on the gating properties of Ca$_V$1.1 currents. The E87A mutation right-shifted the voltage dependence of activation by 12.3 mV, while activation kinetics were not altered (*Figure 5B–G*). In contrast, the E90A mutation accelerated the activation kinetics four- to five-fold and showed a 7.7 mV right-shift of voltage dependence (*Figure 5L–Q*).

Again, MD simulation and MSM analysis reflected these differential functional effects. In accordance with its effect on activation kinetics, the E90A mutation, but not E87A, showed greatly accelerated transition times and reduced energy barriers between the resting and activated states (*Figure 5H–J and R–T*). Furthermore, the free energy maps of all three mutations showed shallower energy wells in the activated states, consistent with their reduced stabilization and their right-shifted voltage dependence of current activation. Also, compared to WT VSD I, the three mutants displayed a decreased drop of the energy minima (ΔG) from resting state 3 to the activated state (*Figure 4L* and *Figure 5J,T*; *Figure 4—figure supplement 2*; *Supplementary file 1*). In the two mutations affecting activation kinetics (E87A/E90A and E90A), resting states 1 and 2 collapsed into a single deep energy well (*Figures 4J* and *5R*), consistent with the notion that in WT VSD I sequential formation of ion-pair

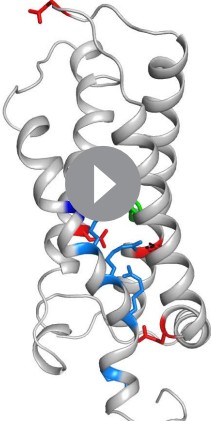

**Video 2.** Movement of the E87A/E90A double mutant of Ca$_V$1.1e voltage sensing domain (VSD) I upon activation and deactivation, highlighting the ion-pair interactions formed between the S4 gating charges (blue) and relevant countercharges (red) in the S2 and S3 helices.
https://elifesciences.org/articles/64087#video2

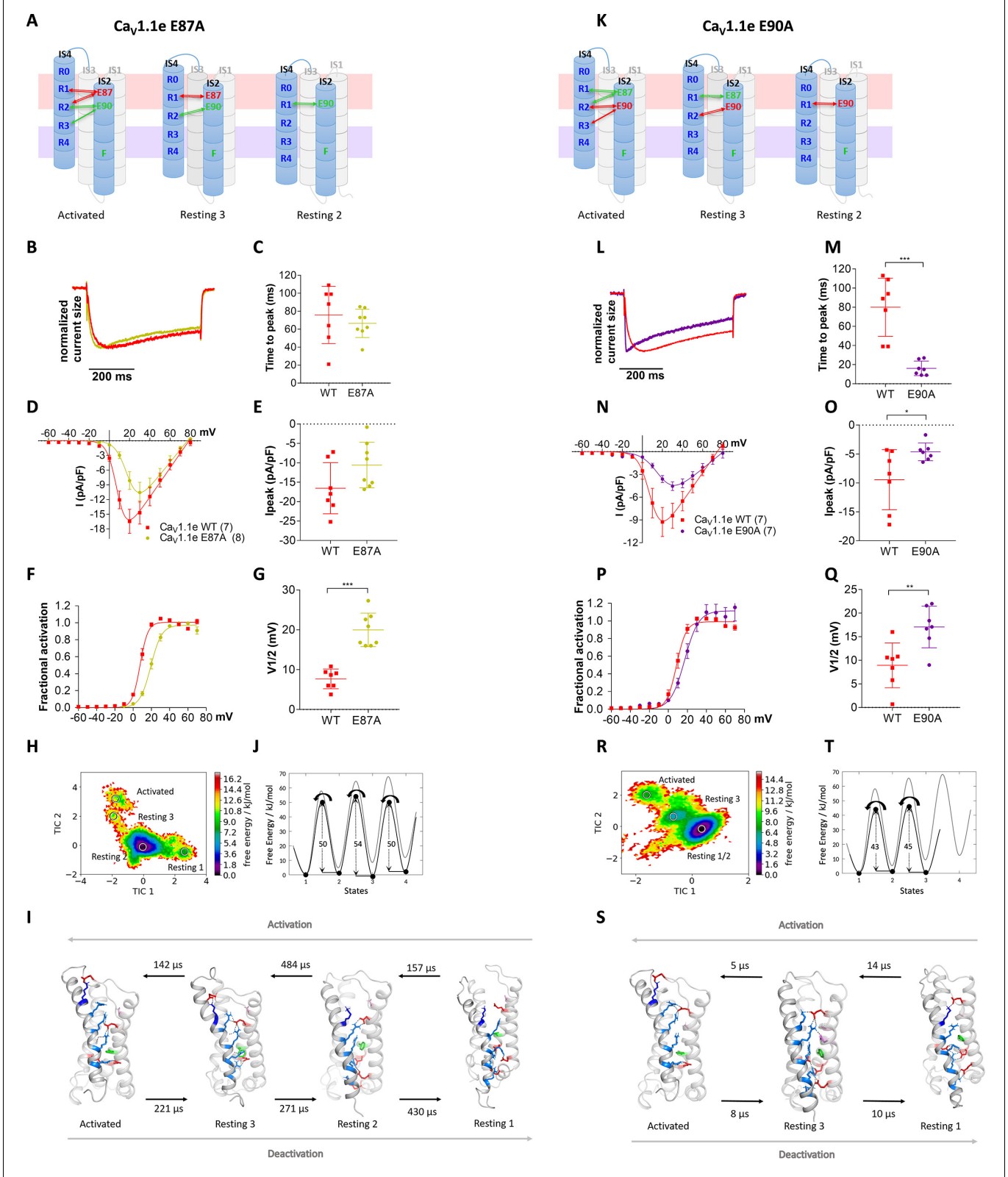

**Figure 5.** Countercharges E87 and E90 differentially regulate kinetics of voltage sensing domain (VSD) I transitions and current activation. (**A,K**) Schematic of VSD I in activated and resting states, showing the loss of ionic interactions upon mutation of E87A or E90A. (**B–G**) In Ca$_V$1.1e E87A right-shifted voltage dependence of activation without affecting kinetics (wildtype [red], E87A [lime]). (**L–Q**) The E90A mutation accelerated kinetics >4-fold and right-shifted voltage dependence of activation (wildtype [red], E87A [purple]). (**B,L**) Normalized representative currents show acceleration of

*Figure 5 continued on next page*

*Figure 5 continued*

activation in E90A (L) but not in E87A (B). (C,M) Time to peak (p=0.47 in C, p=0.00017 in M); (D,N) current-voltage relationship; (E,O) maximum current density (p=0.08 in E, p=0.04 in O); (F,P) voltage dependence of activation; (G,Q) voltage at half-maximal activation (V½) (p=0.000014 in G, p=0.008 in Q). Mean ± SEM; p-values calculated with Student's t-test. (H–J) The time-lagged independent component analysis (tICA) free energy surface and schematic 1D representation of E87A show four macrostates corresponding to resting states 1, 2, 3 and the activated state with energy barriers similar to wildtype (gray) and transition kinetics in the higher µs timescale. (R–T) E90A shows three macrostates corresponding to the resting states 1 and 3 and the activated state, reduced energy barriers, and transition kinetics in the low µs timescale.

The online version of this article includes the following figure supplement(s) for figure 5:

**Figure supplement 1.** Deactivation kinetics are fast in Ca$_V$1.1e wildtype (WT) and in slowly (E87A) and fast-activating (E90A) voltage sensing domain (VSD) I mutants.

interactions between E90 and R1, R2, and R3 is required to stabilize the separate resting states of VSD I, and that the transitions between these states slow down activation kinetics (*Videos 1–4*; *Supplementary file 1*). Thus, the structures derived from our simulations provide mechanistic explanations for how Ca$_V$ channels determine their unique gating properties.

Notably, the differences of the transition kinetics observed in the MSM analysis between VSDs I and IV, and between WT VSD I and the E87A and E90A were manifested in the activating and deactivating direction (*Figures 2*, *4* and *5*). However, patch clamp analysis of deactivation kinetics in WT and mutant VSD I did not reflect these differences (*Figure 5—figure supplement 1*). Upon repolarization to negative membrane potentials, the deactivation time constants of all tested constructs were between 4 and 10 ms and thus near the activation time constants of the fast activating mutants (E87A/E90A and E90A). This is expected considering the distinct dependence of channel activation and deactivation on the actions of multiple VSDs. Upon depolarization, the VSDs need to proceed through all resting states into the activated state before the channel gate will open. Inevitably, the speed of this action is limited by the slowest VSD necessary for channel opening (VSD I in the case of Ca$_V$1.1). In contrast, on deactivation the channel gate closes when the first essential VSD transits from the activated state into resting state 2 (*Figure 5—figure supplement 1*). Principally, this can be any one of the four VSDs. Therefore, channel deactivation will be rapid even if VSD I requires considerably more time to return to its deepest resting state, as predicted by our MSM analysis.

## Discussion

Our advanced structure models of the activated and resting states of Ca$_V$1.1 VSDs fill an important gap in the understanding of the voltage sensing mechanism in eukaryotic voltage-gated cation channels. The results presented in this study demonstrate two surprisingly different VSDs, which substantially differ in the range of S4 helix displacement during activation/deactivation, as well as in the number and position of ionic bonds formed between the outer gating charges and countercharges of the ENC in the resting and activated states. These structural differences correspond to differences in the free energy states and kinetics of the state transitions, which in turn correlate with the experimentally determined kinetics and voltage dependences of current activation. Throughout this structure-function study, the combination of MD simulation and MSM proved to be of exceptionally high predictive value. All tested interaction partners suggested by the model showed the predicted effects on channel gating when experimentally tested by

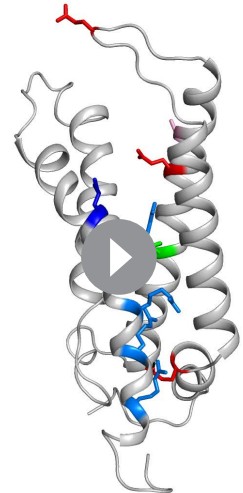

**Video 3.** Movement of the E87A mutant of Ca$_V$1.1e voltage sensing domain (VSD) I upon activation and deactivation, highlighting the ion-pair interactions formed between the S4 gating charges (blue) and relevant countercharges (red) in the S2 and S3 helices.
https://elifesciences.org/articles/64087#video3

mutagenesis and electrophysiology analysis. Conversely, alterations of kinetic properties first recognized experimentally were reliably reproduced and explained by our computer model.

Overall, the structures of the lowest energy states of Ca$_V$1.1 VSDs display the basic features of the sliding helix model (*Catterall et al., 2017*) and of previous simulations of VSD structures (*Tuluc et al., 2016b*; *Yarov-Yarovoy et al., 2012*). Notably, two of the resting states predicted by our models closely correspond to recently described cryo-EM structures of homotetrameric Na$_V$ channels with the VSD in a down-state. Our resting state 1 structure of VSD IVa resembles a mutated Na$_V$Ab captured in the resting state by disulfide locking (Cα RMSD of 1.7 Å) (*Figure 2—figure supplement 4*, *Wisedchaisri et al., 2019*). Our resting state 2 structure of VSD I resembles that of a Na$_V$Ab/Na$_V$1.7 chimera stabilized in a deactivated state by toxin binding (Cα RMSD of 1.9 Å) (*Figure 2—figure supplement 5*, *Xu et al., 2019*). Thus, the voltage sensing action of the eukaryotic Ca$_V$1.1 displays a remarkable similarity to that of its homotetrameric bacterial ancestor. Particularly, the sequential movement of the gating charges across the HCS and the stabilization of the states by their transient formation of ion pairs with negatively charged amino acids of the INC were similarly observed in VSDs I and IV. Evidently, these features represent highly conserved properties of voltage-gated cation channels (prokaryotic and eukaryotic) and probably define the essence of the voltage sensing mechanism.

In contrast, the interactions of the outer gating charges with the negatively charged amino acids of the ENC differed considerably between the two studied VSDs of Ca$_V$1.1, as well as between the two splice variants of VSD IV. Although the outer gating charges of both VSDs established such ion pairs, they used partners in different transmembrane segments (IS2 vs. IVS3). In both VSDs, the examined ion-pair partners of R1 proved to be crucial for stabilizing the voltage sensor in the activated state, as their mutations consistently resulted in a shift of the voltage dependence of activation to more depolarized voltages (*El Ghaleb et al., 2019*; *Tuluc et al., 2016b*). Furthermore, the ion-pair interactions in the two VSDs differed in the extent they were formed in different states. While in VSD IV such ion pairs were restricted to the activated state, in VSD I they were also found in resting states 2 and 3, leading to a substantially stronger stabilization of these resting states in VSD I compared to VSD IV. Concordantly, MSM and site-directed mutagenesis demonstrated that this stabilization of resting states in VSD I causes a dramatic slowing of state transitions and of activation kinetics characteristic for Ca$_V$1.1 currents, respectively. Apparently, the repeated formation and breaking of ionic bonds in consecutive resting states increases the energy barriers between the states and thus slows down the movement of IS4 to the activated state. In contrast, the weaker hydrogen bonds of the outer gating charges of IVS4, or mutation of ion-pair partners in IS3, support fast state transitions and current activation.

This indicates that the negative countercharges in the ENC serve a dual role in the voltage-gating process. They enable the hand-over-hand movement of the gating charges, thus guiding the state transitions of S4 across the membrane electric field, and they stabilize the VSD in the activated states. Accordingly, mutation of specific countercharges differentially affected the voltage dependence and kinetics of activation. While the charge-neutralizing mutation of E87 in IS2, which stabilizes the VSD I in the activated state, perturbed the voltage dependence of activation, mutating E90, which also forms ionic bonds with R1 and R2 in the resting states, accelerated the activation kinetics. MD simulation and MSM suggested that the loss of the stabilizing interactions with the negative countercharge E90 causes the collapse of resting states 1 and 2, thereby enhancing the speed of VSD I transition into the activated state. As E90 also contributed to stabilizing the activated state, its mutation also caused a right-shift of the voltage dependence of activation. The importance of activated-state-stabilizing interactions of the outermost gating charges for specifically setting the voltage dependence of activation is consistent with earlier mutagenesis experiments of R1 in VSD I and of R1, R2, and D1196 in VSD IV, all of which right-shifted voltage dependence without changing activation kinetics (*El Ghaleb et al., 2019*; *Tuluc et al., 2016b*). Unlike VSD I, VSD IV establishes stabilizing ion pairs exclusively in the activated state. The lack of stabilizing ion pairs in the resting states is consistent with the intrinsically fast activation kinetics of this VSD. However as in VSD I, weakening the countercharges in IVS3 either physiologically, by insertion of exon 29 in the IVS3-S4 linker, or experimentally, by mutagenesis, caused a right-shift of the voltage dependence of activation with little effect on kinetics (*Tuluc et al., 2016b*). Thus, ion-pair formation of R1 with E87 and E90 in VSD I, and of R1 in VSD IV with D1196 are functionally equivalent in stabilizing the activated state in both VSDs, whereas ion-pair formation of multiple gating charges with E90 that

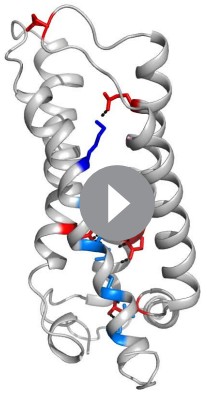

**Video 4.** Movement of the E90A mutant of Ca$_V$1.1e voltage sensing domain (VSD) I upon activation and deactivation, highlighting the ion-pair interactions formed between the S4 gating charges (blue) and relevant countercharges (red) in the S2 and S3 helices. https://elifesciences.org/articles/64087#video4

stabilize the resting states is specific for VSD I and represents the structural correlate of Ca$_V$1.1 slow activation.

If the outward motion of the S4 helix with the sequential stabilization of the inner gating charges by ion-pair formation with the INC is the general theme of the voltage sensing process, the interactions between the outer gating charges with countercharges of the ENC represent the variations to this theme. They differ in number, strength, and position from analogous interactions described in the structures of other channels, and even between the VSDs of a single channel. Importantly, in which states they are established determines which biophysical properties of channel gating are being modulated. The general principle derived from our modeling and mutagenesis experiments is that stabilization of VSDs in the activated state supports channel opening by shifting the voltage dependence of activation to more hyperpolarizing potentials. On the other hand, ion-bond formation between the gating charges and the ENC in resting states delays channel opening by slowing down the activation kinetics. Interestingly, at least in Ca$_V$1.1, these actions are divided between separate VSDs. Thus, our findings provide a molecular mechanism explaining how channels using the same general voltage sensing mechanism can produce very distinct gating properties and how in pseudo-tetrameric eukaryotic voltage-gated cation channels the distinct VSDs cooperate in establishing the characteristic gating properties.

## Materials and methods

### Key resources table

| Reagent type (species) or resource | Designation | Source or reference | Identifiers | Additional information |
|---|---|---|---|---|
| Gene (human) | *CACNA1S* | *Wu et al., 2016* | Q13698 | |
| Gene (rabbit) | *CACNA1S* | *Grabner et al., 1998* | P07293 | |
| Cell line (mouse) | GLT, dysgenic skeletal myotubes | *Powell et al., 1996* | GLT; mdg/mdg | Ca$_V$1.1-null |
| Transfected construct (rabbit) | GFP-Ca$_V$1.1e (wild type) | *Tuluc et al., 2009* | | Ca$_V$1.1-D exon 29 |
| Transfected construct (rabbit) | GFP-Ca$_V$1.1e -E87A/E90A, -E87A, -E90A | This paper | | |
| Antibody | Rabbit polyclonal anti-GFP | Invitrogen Thermo Fisher | A-6455, RRID:AB_221570 | IF (1:10,000) |
| Antibody | Mouse monoclonal anti-RyR | Invitrogen Thermo Fisher | (MA3-925) 34 C RRID: AB_2254138 | IF (1:500) |
| Software, algorithm | AMBER simulation software | AmberMD | RRID: SCR_014230 | |
| Software, algorithm | AmberTools 19 | AmberMD | RRID: SCR_018497 | |
| Software, algorithm | PyMOL | Schrödinger | RRID: SCR_000305 | |
| Software, algorithm | Clampex | Clampex | Version 10.2 RRID:SCR_011323 | |
| Software, algorithm | Clampfit | Clampfit | Version 10.7 RRID:SCR_011323 | |
| Software, algorithm | SigmaPlot | SigmaPlot | Version 12.0 RRID:SCR_003210 | |
| Software, algorithm | GraphPad Prism | GraphPad Prism | Version 7 RRID:SCR_002798 | |

## Homology model of the Ca$_V$1.1 $\alpha_1$ subunit

We predicted the structure of the human WT Ca$_V$1.1 $\alpha_1$ subunit by making a homology model based on the cryo-EM structure of the rabbit Ca$_V$1.1 $\alpha_1$ subunit with the VSDs in the up-state and the pore closed (*Wu et al., 2016*). Homology modeling has been performed using Rosetta and MOE (Molecular Operating Environment, version 2018.08, Molecular Computing Group Inc, Montreal, Canada). The sequence identity between the rabbit and the human Ca$_V$1.1 $\alpha$1 subunit is 92.6%, the sequence similarity even 95.6%. Because of the high sequence similarity and identity between the human and the rabbit Ca$_V$1.1, we generated only 10 homology models and chose the one model with the best energy score as starting structure for further minimizations, equilibrations, and simulations. The fragment-based cyclic coordinate descent algorithm implemented in Rosetta was used to generate structures for loops that were not resolved in the Ca$_V$1.1 $\alpha_1$ subunit template (*Supplementary file 1*; input scripts – IS1, *Canutescu and Dunbrack, 2003*; *Wang et al., 2007*). The C-terminal and N-terminal parts of each domain were capped with acetylamide and *N*-methylamide to avoid perturbations by free charged functional groups. The structure model was embedded in a plasma membrane consisting of POPC (1-palmitoyl2-oleoyl-sn-glycero-3-phosphocholine) and cholesterol in a 3:1 ratio, using the CHARMM-GUI Membrane Builder (*Jo et al., 2009*). Water molecules and 0.15 M NaCl were included in the simulation box. Energy minimizations of the WT and the mutants in the membrane environment were performed. The topology was generated with the tleap tool of the Amber-Tools18 (*Case et al., 2018*), using force fields for proteins and lipids, ff14SBonlysc and Lipid14 (*Dickson et al., 2014*), respectively. The WT and mutant structures were heated from 0 to 300 K in two steps, keeping the lipids fixed, and then equilibrated over 1 ns. Then MD simulations were performed for 10 ns, with time steps of 2 fs, at 300 K and in anisotropic pressure scaling conditions as suitable for membrane proteins. Van der Waals and short-range electrostatic interactions were cut off at 10 Å, whereas long-range electrostatics were calculated by the particle mesh Ewald method. A hierarchical clustering was performed on the 10 ns trajectory using an RMSD distance cut-off criterion of 2.5 Å, resulting in three clusters. We chose the highest populated cluster representative for all further steps. PyMOL Molecular Graphics System was used to visualize the key interactions and point out differences in the WT and mutant structures (PyMOL Molecular Graphics System, version 2.0, Schrödinger, LLC).

## Enhanced sampling and MD simulation protocol

Because high-resolution structures of resting states of Ca$_V$ and Na$_V$ channels are still lacking, we applied MD simulation and MSM of individual VSD in the context of the whole channel to predict the structures and energy levels of resting states (*Chodera and Noé, 2014*). The workflow of the modeling procedure is summarized in *Figure 2—figure supplement 1*. To overcome the high energy barriers and the timescale limitations of MD simulations, we applied Umbrella sampling as enhanced sampling technique. As collective variable we used the distance between the S4 gating charge residues (R1, R2, and R3) and anchor residues at the intracellular helical ends of the VSDs located in S1 and S3, by using a force constant of the harmonic spring potential of 80 kcal/mol*Å$^2$ to pull the S4 helix downward. Starting from the equilibrated structure, the Umbrella windows decreased between a distance of 24.0–14.0 Å using a step size of 1 Å. Each Umbrella window was simulated for 100 ns. After 20 ns of simulation time, the current conformation was extracted and used as starting structures for the next Umbrella window. The force constant of 80 kcal/mol*Å$^2$ was determined to allow a sliding movement of S4 with minimal distortion of the VSDs. Additionally, we applied a weak backbone restraint on the $\varphi$ torsion angle of the S4 helix of 50 kcal/mol*rad$^2$ to guarantee a minimum of local artifacts of the Umbrella sampling process, that is, loss of secondary structure of the S4 helix. This combination of pulling and torsional restraint was tested and resulted in a sliding movement of the S4 helix without observing unfolding events. Note that the combination of restraints and Umbrella sampling does not result in equilibrium distributions, due to insufficient overlap between the individual sampling windows. Rather the Umbrella sampling was applied to generate conformations along a potential deactivation pathway; however, no states were pre-defined based on the Umbrella sampling. Hence, the Umbrella sampling was used as a mechanical force to pull the S4 helix in the absence of a membrane potential. To obtain the different activation and resting states, we used the resulting pathway of the combined Umbrella sampling trajectories and

clustered it using a small distance cutoff criterion to also obtain cluster representatives at transition state regions. Using this procedure, we cannot exclude the possibility of other substantially different pathways (e.g., such that involve helix rotation and formation or breaking of interactions before or after S4 translocation). However, from our calculations we see no indications of the existence of such completely different pathways, which are kinetically accessible. Thus, to reconstruct the transition kinetics and to improve the sampling efficiency, we clustered the Umbrella sampling trajectories applying the program implemented in the AMBER suite *cpptraj* (*Roe and Cheatham, 2013*) by using the average linkage hierarchical clustering algorithm with an RMSD distance cutoff criterion of 1.2 Å resulting in a large number of clusters. The choice of the distance cutoff is optimized to obtain a broad cluster distribution within the conformational space of each VSD. The cluster representatives of the different activation states were equilibrated and simulated for 100 ns using the AMBER18 simulation package. For the resulting trajectories, a tICA was performed using the Python library PyEMMA 2 employing a lag time of 10 ns.

Construction of tICA and Markov state models is a dimensionality reduction technique, detecting the slowest relaxing degrees of freedom and facilitating the kinetic clustering, which is crucial for building an MSM (*Pérez-Hernández and Noé, 2016*). It linearly transforms a set of high-dimensional input coordinates to a set of output coordinates, by finding a subspace of good reaction coordinates. Thereby, tICA finds coordinates of maximal autocorrelation at a given lag time. The lag time sets a lower limit to the timescales considered in the tICA and the MSM.

Thermodynamics and kinetics were calculated with a Markov state model by using PyEMMA 2 (*Scherer et al., 2015*), which uses the k-means clustering algorithm to define microstates and the Perron Cluster Cluster Analysis (PCCA)+ clustering algorithm to coarse-grain the microstates to macrostates (*Röblitz and Weber, 2013*; *Wu and Noé, 2020*). The k-means clustering represents an iterative and robust clustering algorithm, which partitions the dataset into pre-defined distinct non-overlapping clusters, with the aim to make the intra-cluster points as similar as possible and keeping the subgroups as different as possible.

To construct coarse-grained models, the PCCA uses the eigenspectrum of a transition probability matrix. The eigenvectors corresponding to the Perron Cluster can be further used to coarse-grain an MSM. Here, we applied the PCCA+ clustering, as it is more robust than PCCA. PCCA+ tries to identify a set of indicator functions that can reproduce the slowest dynamical eigenvectors. PCCA+ relies on a maximum likelihood estimate of the transition.

To build the Markov state model, we used the Cα coordinates of the respective S4 transmembrane helix, defined 100 microstates using the k-means clustering algorithm and applied a lag time of 10 ns. The sampling efficiency and the reliability of the Markov state model (e.g., defining optimal feature mappings) can be evaluated with the Chapman-Kolmogorov test (*Figure 2—figure supplement 4*), by using the variational approach for Markov processes and by taking into account the fraction of states used, as the network states must be fully connected to calculate probabilities of transitions and the relative equilibrium probabilities (*Likas et al., 2003*). The construction of the MSM allows to quantify thermodynamic and kinetic properties of the resulting ensembles without the intrinsic bias resulting from the seeding process (*Figure 2—figure supplement 1*). The first stage of the MSM is to discretize the obtained conformational space into the so-called microstates, grouping together conformations of the system that can exchange rapidly (e.g., by k-means clustering). The aim is to construct a kinetically relevant clustering by using a geometric criterion, which still allows a quantitative connection with experiments, due to their high resolution. To identify the kinetic relevance of the clustering, an appropriate lag time, that is, observation interval, has to be chosen. This resulting microstate model can then be used as starting point for a kinetic clustering. To create a more understandable model, a kinetic clustering of a relevant set of microstates to the so-called macrostates can be performed, which are larger aggregates that correspond to the free energy wells (e.g., by PCCA+ clustering). The additional kinetic clustering into macrostates results in a more compact representation than the microstate model and thereby allows an easier processing and understanding of the conformational space. Thus, these qualitative models are ideal for generating new hypotheses, which can then be tested again with higher resolution models and experiments. The MSM was constructed by following the guidelines and input commands from the provided tutorial (http://www.emma-project.org/latest/tutorial.html#jupyter-notebook-tutorials).

To calculate the 1D free energy barriers $K^{\ddagger}$ we used the obtained mean first passage times k from the MSM and calculated the barriers according to the transition state theory with the following equation:

$$k = \frac{k_B T}{h} * K^{\ddagger}$$

## Expression plasmids

Cloning procedure for GFP-Ca$_V$1.1e WT was previously described (*Tuluc et al., 2009*). For better comparison with the literature, the non-corrected version of Ca$_V$1.1 was used. This Ca$_V$1.1 construct contains a lysine in position R1 of the VSD I, which results in a 12 mV left-shifted V½ compared to the construct with the evolutionary conserved arginine in position R1 (*El Ghaleb et al., 2019*). To generate the double mutant GFP-Ca$_V$1.1e-E87A/E90A and the single mutants GFP-Ca$_V$1.1e-E87A and GFP-Ca$_V$1.1e-E90A, aa E87 and E90 were neutralized by SOE-PCR (*Supplementary file 5*). Briefly, nt 1–1113 of the coding sequence of Ca$_V$1.1e (nt 226–1338 of *CACNA1S* NCBI reference sequence NM_001101720.1) were PCR-amplified with overlapping primers introducing the point mutation A > C at position nt 260 and/or the point mutation A > G at position nt 269 (nt 485 and nt 494, respectively, of NM_001101720.1) in separate PCR reactions using GFP-Ca$_V$1.1e-WT as template. The two separate PCR products were then used as templates for a final PCR reaction with flanking primers to connect the nucleotide sequences. This fragment was then SalI/EcoRI digested and cloned into the respective sites of GFP-Ca$_V$1.1e WT. Sequence integrity of the newly generated constructs was confirmed by sequencing (MWG Biotech, Martinsried, Germany).

## Cell culture and transfection

Myoblasts of the dysgenic (mdg/mdg) cell line GLT were cultured as previously described in *Powell et al., 1996*. Briefly, cells were plated on 35 mm culture dishes and transfected with 0.5 μg of the desired Ca$_V$1 subunit 4 days after plating using FuGENE-HD transfection reagent (Promega). After 7–8 days in culture, transfected myotubes showing GFP fluorescence were analyzed by electrophysiology or fixed for immunolabeling after 9–10 days in culture. The auxiliary calcium channel subunits α2δ−1, β1a, and γ$_1$, along with the STAC3 protein and ryanodine receptor, are endogenously expressed by GLT myotubes, enabling functional membrane incorporation of the channel constructs in the triad junction.

## Immunofluorescence and antibodies

Paraformaldehyde-fixed cultures were immunolabeled as previously described (*Flucher et al., 1994*) with rabbit polyclonal anti-GFP (1:10,000; Invitrogen Thermo Fisher) and mouse monoclonal anti-RyR (34 C; 1:500; Invitrogen Thermo Fisher) and fluorescently labeled with goat anti-rabbit Alexa-488 and secondary goat anti-mouse Alexa-594 (1:4000), respectively. Thus, the anti-GFP label and the intrinsic GFP signal were both recorded in the green channel. Samples were observed using a 60×, 1.42 NA objective with a BX53 Olympus microscope and 14-bit images were captured with a cooled charge-coupled device camera (XM10, Olympus) and CellSens Dimension image-processing software (Olympus). Image composites were arranged in Adobe Photoshop CS6 (Adobe Systems Inc) and linear adjustments were performed to correct black level and contrast.

## Electrophysiology and data analysis

Calcium currents were recorded with the whole-cell patch clamp technique in voltage clamp mode using an Axopatch 200A amplifier (Axon Instruments). Patch pipettes (borosilicate glass; Science Products) had resistances between 1.5 and 3.5 MΩ when filled with (mM) 145 Cs-aspartate, 2 MgCl$_2$, 10 HEPES, 0.1 Cs-EGTA, and 2 Mg-ATP (pH 7.4 with CsOH). The extracellular bath solution contained (mM) 10 CaCl$_2$, 145 tetraethylammonium chloride, and 10 HEPES (pH 7.4 with tetra-ethylammonium hydroxide). Data acquisition and command potentials were controlled by pCLAMP software (Clampex version 10.2; Axon Instruments); analysis was performed using Clampfit 10.7 (Axon Instruments) and SigmaPlot 12.0 (SPSS Science) software. The current-voltage dependence was fitted according to

$$I = G_{max} * (V - V_{rev})/(1 + \exp(-(V - V_{1/2})/k))$$

where $G_{max}$ is the maximum conductance of the L-type calcium currents, $V_{rev}$ is the extrapolated

reversal potential of the calcium current, $V_{1/2}$ is the potential for half-maximal conductance, and k is the slope. The conductance was calculated using G = (− I * 1000)/($V_{rev}$ − V), and its voltage dependence was fitted according to a Boltzmann distribution:

$$G = G_{max}/(1 + \exp(-(V - V_{1/2})/k))$$

## Statistical analysis

All four experimental groups were analyzed in transiently transfected cells from three to five independent cell passages. The E87A/E90A, E87A, and E90A variants of $Ca_V1.1e$ were always recorded in parallel with the WT $Ca_V1.1e$ in cells of the same passage to obtain the best controls for statistical comparison. Consequently, the values for WT controls vary slightly between conditions. The means, standard errors (SE), and p-values were calculated using the Student's t-test, two-tailed, with significance criteria *$p<0.05$, **$p<0.01$, and ***$p<0.001$. Two-way repeated measures ANOVA, with the Holm Sidak post hoc test, was used to calculate p-values of deactivation.

## Acknowledgements

We thank I Mahlknecht, N Kranebitter, M Bacher, and S Demetz for excellent technical support. The computational data have been obtained in part using the Vienna Scientific Cluster (VSC).

## Additional information

### Funding

| Funder | Grant reference number | Author |
| --- | --- | --- |
| Austrian Science Fund | P30402 | Bernhard E Flucher |
| Austrian Science Fund | DOC30 | Bernhard E Flucher |
| Austrian Science Fund | T855 | Marta Campiglio |

The funders had no role in study design, data collection and interpretation, or the decision to submit the work for publication.

### Author contributions

Monica L Fernández-Quintero, Yousra El Ghaleb, Conceptualization, Data curation, Formal analysis, Investigation, Visualization, Methodology, Writing - review and editing; Petronel Tuluc, Conceptualization, Formal analysis, Supervision, Writing - review and editing; Marta Campiglio, Supervision, Methodology; Klaus R Liedl, Resources, Supervision, Methodology; Bernhard E Flucher, Conceptualization, Resources, Supervision, Funding acquisition, Writing - original draft, Project administration, Writing - review and editing

### Author ORCIDs

Monica L Fernández-Quintero ⬤ https://orcid.org/0000-0002-6811-6283
Yousra El Ghaleb ⬤ https://orcid.org/0000-0002-0829-5865
Petronel Tuluc ⬤ https://orcid.org/0000-0003-3660-6138
Marta Campiglio ⬤ http://orcid.org/0000-0002-9629-2073
Klaus R Liedl ⬤ http://orcid.org/0000-0002-0985-2299
Bernhard E Flucher ⬤ https://orcid.org/0000-0002-5255-4705

### Decision letter and Author response

Decision letter https://doi.org/10.7554/eLife.64087.sa1
Author response https://doi.org/10.7554/eLife.64087.sa2

## Additional files

### Supplementary files

• Supplementary file 1. Mean values and confidence intervals for probabilities and mean first passage times calculated from the Bayesian Markov state modeling (MSM), and the confidence intervals are calculated at a confidence level of 95%.

• Supplementary file 2. Current properties of E87A/E90A, E87A, E90A, and wildtype (WT) controls.

• Supplementary file 3. Linear interaction energy (LIE) calculations with the program *cpptraj* for the two voltage sensing domains (VSDs) and the mutants to calculate the electrostatic interactions of the S4 helix with all other parts of the voltage sensor.

• Supplementary file 4. Summary table of the modeled loops in the voltage sensors with ab initio Rosetta (Robetta).

• Supplementary file 5. Primers used for site-directed mutagenesis.

• Supplementary file 6. Input scripts.

• Transparent reporting form

### Data availability

All data generated or analysed during this study are included in the manuscript and supporting files. The pdb structures of the models of the activated and the resting states of both the WT VSDs and the mutants are available from the Dryad server https://doi.org/10.5061/dryad.hhmgqnkfd.

The following dataset was generated:

| Author(s) | Year | Dataset title | Dataset URL | Database and Identifier |
|---|---|---|---|---|
| Fernández-Quintero ML, El Ghaleb Y, Tuluc P, Campiglio M, Liedl KR, Flucher BE | 2020 | Structural determinants of voltage-gating properties in calcium channels | https://doi.org/10.5061/dryad.hhmgqnkfd | Dryad Digital Repository, 10.5061/dryad.hhmgqnkfd |

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
