## [Decision Letter]

**Acceptance summary:**

Here Fernandez-Quintero et al demonstrate how changes in acidic residues in the voltage sensors of skeletal muscle Cav1.1 calcium channels can control their function. The atomistic simulations and Markov State Model developed here describe voltage-sensor movements from resting to activated states, revealing distinct kinetic barriers in different domains. This is backed by mutagenesis studies which show differentially regulated channel function for mutants of one domain that controls slow activation. The results provide new insight into voltage-gated ion channel function.

**Decision letter after peer review:**

Thank you for submitting your article "Structural determinants of voltage-gating properties in calcium channels" for consideration by *eLife*. Your article has been reviewed by 3 peer reviewers, one of whom is a member of our Board of Reviewing Editors, and the evaluation has been overseen by Richard Aldrich as the Senior Editor. The following individual involved in review of your submission has agreed to reveal their identity: Vladimir Yarov-Yarovoy (Reviewer #2).

The reviewers have discussed the reviews with one another and the Reviewing Editor has drafted this decision to help you prepare a revised submission.

Summary:

Your manuscript has been evaluated by three reviewers who find the results interesting. The models demonstrate how changes in the acidic residues in the voltage sensors of skeletal muscle Ca_V_1.1 may control kinetics and voltage dependence. The molecular models of VSDs I and IV describe a sliding helix mechanism, but with different activation barriers, such that VSD I appears to control slow activation of Ca_V_1.1.

Essential revisions:

The reviewers are concerned about the heavy reliance on simulations that are not well described and lack measures of reliability. Better description of the conceptual basis of the methods for non-MD expects and much more precise definition of the methods for MD-experts are needed so that the readers can accept that the models are reasonable representations of the gating motions studied.

The reviewers criticise the lack of statistical reliability tests, with absence of measures of error, convergence and reproducibility, each of which are required for publication in *eLife*. This is particularly important as trajectories of the order of 100 ns need to represent events that actually take place on 100 microsecond or longer time scales in physiology, such that simulations could be merely anecdotal, incomplete, dependent on starting models or dependent on the choices made in enhanced sampling techniques. Currently, the description leaves the reader to guess what has been done, unsure of the impacts those decisions might have on the findings, as well as the strengths and weaknesses of the approaches. Thus, as well as detailed responses to the comments of the reviewers, it is essential that the simulation methods be better described and justified, with results supplemented by rigorous proofs of sampling (e.g. interconversion, back and forth, between identified macrostates), convergence (e.g. MSM results changing to within a small tolerance) and reproducibility (e.g. not being dependent on the selected initial Umbrella Sampling), along with quantitative comparisons to experiment.

Reviewer #1:

This manuscript represents a study of the voltage sensor movements in homology models of Ca_V_1.1 to explain its slow kinetics and right shifted V dependence in terms of the roles of distinct structures and ion pair interactions in VSDs I and IV, backed by electrophysiology. In particular, the results suggest interactions specific to subunit chemistry can drastically affect activation. This is backed by mutagenesis and electrophysiology seeing special roles played by residues, such as E87 affecting V dependence while E90 affects rate, representing important information on how subunits of a Ca_V_ control voltage gating. The study is well written and presents a unified story to explain kinetics and V dependence. The models are used in MD simulations to examine thermodynamics and kinetics of the voltage-sensing transitions, but they are not well described and reliability tests appear non-existent, as discussed more below. A more involved examination of sampling, reproducibility and errors is needed to trust those results. Without this it is hard to judge if they are even qualitatively consistent with the electrophysiology to provide insight into voltage sensing.

Although given a fairly wordy description in the methods, the MD approaches are not well described and I am left unsure of what was actually done. Umbrella sampling was first used to "overcome high barriers". But details of the Umbrella Sampling are not described. I assume this was 1D sampling, but what is the reaction coordinate? On page 18 the authors talk about "pulling S4 down" with K = 80 kcal/mol*rad*2. What does that even mean having an angular restraint? Likewise, the authors mention some torsion restraint on S4, but it is not clear what it was actually acting on and its purpose. Was this the restraint for each window? How many windows and how was the coordinate defined? Was each of many windows run for 100ns or was that just subsequent sampling in a few macrostates? Did the windows overlap well and was the preliminary free energy profile from this Umbrella Sampling (not provided) converged? As this Umbrella Sampling set the stage for all of the MD results, how it was carried out is critical. Importantly, as the authors admit the barriers are high and timescales long for S4 movements, if not captured adequately by the chosen reaction coordinate may lead to far-from-equilibrium starting points for subsequent simulations, tICA analysis and MSM. (n.b. The outward sliding motion of S4 is important for analysis and interpretation, and while this may be the case, until we know how the starting structures were made with US we cannot tell if it is an artifact of the starting trajectory or not.)

The authors then took clusters and simulated different "activation states" for 100ns each. Does that mean activated states, or activation states between activated and resting states? How were these states actually identified? We would need to see those defined states from the Umbrella simulations and how they were chosen. Are final results depending on such decisions?

The methods description following this stage is vague. The authors say on page 19 that they "discretize the obtained conformational space into so-called microstates, grouping together conformations of the system that can exchange rapidly (e.g., k-means clustering)." Most *eLife* readers will not understand this, and it is vague in that it does not specify the actual method, but some example method "k-means clustering" without explaining what it means. They then say "To create a more understandable model, a kinetic clustering of a relevant set of microstates to so-called macrostates, can be performed, which are larger aggregates that correspond to the free energy wells (e.g., PCCA+ clustering)." Again stating an example method called PCCA+ clustering. Was this what was used, and what does it mean? Then, MSM is used to extract thermodynamics and kinetics. Details of this analysis, its reliability and reproducibility are completely lacking.

In Figure 2 and later figures, the authors choose to plot sampling and identify states in the dominant two tICA vectors. What do these vectors actually mean physically? How do we know only 2 such order parameters characterise the slowest coordinates in the activation transition? Subsequent panels in Figure 2 suggest an activation/deactivation reaction coordinate that is visualised as a sliding S4, presumably due to changing structures in the macrostates. How does concerted movement between minima in tICA1,2 translate to this S4 sliding movement? Those two order parameters would need visualisation to become meaningful to the reader.

Throughout the paper the data is presented without any errors. e.g. What are the error bars in transition mean first passage times in Figure 2? This is particularly important for barriers and the exponentially sensitive rates. Figure 2d suggests activation barriers of 60 kcal/mol. Is that even possible? If one were to estimate a rate for hopping over such barriers, it would not be microseconds (more likely millennia)! How reliable are those barriers (and estimated rates)? Analysis to calculate errors and prove evidence for reproducibility of the results is missing.

Regarding these barriers, I have concerns about sampling of interconversion of macrostates to capture the MFPT. e.g. Looking at Figure 2a, it seems that there is no sampling between them in the chosen TICA1-2 space, and this raises doubt about the free energies and rates. Regarding apparent lack of overlap between sampling in macrostates (most apparent in Figure 2A), what is the origin of this? This may come down to the Umbrella Sampling used to get starting points for subsequent 100ns simulations for MSM analysis. Some reaction coordinate (unspecified) was presumed for Umbrella Sampling and pulling along that direction may have yielded patches of sampling such that the minima found in the MSM are artifacts of the starting Umbrella trajectory (sequence of windows). Would the same results emerge from a different initial starting method?

In Figure 2 panel A, the scale goes to 10kT whereas the predicted barriers are much higher (60 kcal/mol). These are given different units, and 60 kcal/mol is of order 100kT, being on a different order to the scale of the free energy maps. n.b. On page 9 the authors now refer to barriers of 50 kJ/mol (a third different unit of energy and again different magnitude). This of order 10-15 kcal/mol and appears inconsistent with the above data. Units again on page 10 in kJ/mol appear wrong. I do not trust the predictions based on the data shown. We would need to be convinced of good sampling of transitions both back and forth between macrostates, such that the MFPT estimates have high accuracy, with propagated errors to rates estimated. Regarding rate estimates: How is it that the predicted barriers between successive macrostates change by 10 kcal/mol (e.g. sequence of barriers in panel D), but rates between macrostates (e.g. in panel B) remain of the same order of magnitude (1ms)? This makes no sense to me. Perhaps the barriers in panel D and other similar graphs are not based on any calculated data?

The authors write that "although the absolute times derived from MSM may not correspond to the actual transition times of the VSD upon physiological activation and deactivation, relative differences between transition times provide meaningful information when compared between different VSDs or functionally different mutants". Sources of error in absolute times (and comparison to known experimental times) is required. Regarding reliability of relative rates, how do we know to believe the above statement? Besides model and force field dependence, proof of error and reproducibility of what was sampled is needed.

On page 11 the authors state that "additionally formed ion-pairs in the resting

states of VSD I are paralleled by a remarkable increase in the energy barriers and the state transition times, suggesting that the number and strength of interactions between the gating charges and the ENC transiently formed in the resting states determine the slow activation kinetics of VSD I." It is suggested here that those changed interactions change the barriers, as indicated in Fig2D-H by 20 kcal/mol. Increased barrier like that means a prefactor (assuming similar order of friction) of exp(-20kcal/mol / k_B_ T) of the order of 10^-14^. Yet the rates change between just micro and millisecond. Again, perhaps the representation of the barriers in the figures are not actual computed barriers? Related, on page 13 the authors write that their result "substantiates the reliability and predictive value of the kinetic analysis of our MD simulation". A much more quantitative comparison than this is needed to demonstrate reliability.

Reviewer #2:

Manuscript by Monica Fernández-Quintero et al. describes study of molecular mechanism of voltage-gated calcium channel gating using experimental and computational modeling approaches. Authors identified specific ion-pair interactions stabilizing either activated or resting states of voltage sensing domains I and IV in eukaryotic voltage-gated calcium channel. Overall, the results are promising and provide evidence towards evolutionary functional conservation of the sliding helix hypothesis. This study provides key structural insights into voltage-dependence and kinetics of current activation of eukaryotic voltage-gated calcium channels.

1. Discuss potential role of other interactions involving hydrophobic and polar residues in stabilizing specific VSD states besides the ion-pair interactions identified in this study. Also note that residue size and hydrophobicity may also contribute to differences in VSD kinetics – see Jérôme J. Lacroix et al. (2014) Moving gating charges through the gating pore in a K_V_ channel voltage sensor. Proc Natl Acad Sci U S A, 111(19): E1950-9. doi: 10.1073/pnas.1406161111.

2. A large part of the study is computational modeling to generated models of VSDs in different resting states and run MD simulations to compute transition kinetics. However, to prove the rigor of the approach authors need to describe essential details and related results in supplemental figures and/or text. Specifically:

a. What was the reaction coordinate in the Umbrella sampling?

b. How many windows used and how well they overlapped?

c. How did the authors decide if the Umbrella sampling has generated enough data for the following steps?

d. The author mentioned for clustering the Umbrella sampling trajectory "… resulting in a large number of clusters". How many clusters exactly, given all of these clusters were subsequently simulated for 100 ns?

e. How was the time lag of 10 ns chosen for the tlCA analysis in the MSM modeling?

f. The authors should show an implied timescales (ITS) plot as a function of different lag times. Was it convergent?

g. The authors should also show the results of the Chapman-Kolmogorov test to ensure the dynamics observed is Markovian so that the results can be meaningful.

3. Discuss accuracy of absolute activation and deactivation times estimated by MSM – how far away they are from the actual transition times observed experimentally?

4. An assumption in the computational modeling part of this study is that transitions between VSD states can happen at equilibrium. However, this rarely happens in reality. The activation or deactivation can be considered as a non-equilibrium process by which the electric field exerts force on gating charges to trigger conformational changes in VSDs. Authors should discuss strength and weaknesses of chosen computational modeling methods.

5. Note that while the experimental testing is in agreement with observations form computations models, it is not a direct validation of accuracy of the computational prediction. Given the largely unchanged conformation of the rest of the channel except S4, the computational modeling at its best can only infer S4 gating charges movement in individual VSD in a specific physiological state of the channel. The experiments, on the other hand, only showed macroscopic view of the channel activation, in which the contribution of the gating charges movement, VSDs cooperation, and pore coupling is complex. The authors need to discuss these limitations.

6. The authors studied activation and deactivation kinetics of the gating charges movement using the computational modeling approaches. Discuss why only activation kinetics have been studied experimentally.

7. Can authors provide computational ion-pair energy calculations to support VSD state-stabilization hypotheses?

8. Provide in the Supplementary Information example scripts and command lines used for Rosetta homology modeling, MD simulations, and MSM calculations. Such scripts should contain enough information such that someone else could independently run and validate author's data.

9. Provide confidence intervals for experimental data.

10. Provide PDB coordinates of Rosetta Ca_V_1.1 models as text files in the Supplementary Information.

Reviewer #3:

Fernandez-Quintero et al. show how changes in the negatively charged amino acid residues in the voltage sensor of skeletal muscle Ca_V_1.1 calcium channels can control their function. These calcium channels serve two physiological functions with different kinetics and voltage dependence: excitation-contraction coupling triggered by conformational coupling with the ryanodine receptor in the sarcoplasmic reticulum and calcium entry triggered by opening of their pore. Voltage-driven movement of the S4 segments in their voltage sensors initiate both functions. Molecular models are developed here for the function of VSD I and VSD IV describing the transmembrane movements of their S4 segments from resting to activated states. In each case, the molecular models fit a sliding helix mechanism of VSD function in which the S4 segments exchange ion pair partners between the extracellular negative cluster and the intracellular negative cluster of counterions in the voltage sensor. The kinetic barriers for movement of the S4 segment in VSD IV were much lower than VSD I, suggesting that VSD I controls the slow activation of the calcium conductance of Ca_V_1.1 channels. Introducing mutations in two negatively charged amino acid residues in the extracellular negative cluster of VSD I (E87A and E90A) showed that they differentially regulate channel function. E87A positively shifted the voltage dependence of activation, whereas E90A gave a smaller positive shift in activation and slowed the rate of activation by four-fold.

Understanding how molecular specializations in VSDs control ion channel function is an important goal of molecular physiology. Because of its unique functional properties, the Ca_V_1.1 channel provides an interesting experimental model. The results presented here are an important advance, as they provide a detailed new molecular model of VSD function and they show how a conserved molecular feature of VSDs, the negatively charged amino acid residues in the extracellular negative cluster, control the voltage dependence and rate of activation of VSD I.

Specific Comments:

Line 33. It is an overstatement to say calcium channels control most functions in excitable cells. Many hormone receptors and other regulators act independently of them. It seems more accurate to say that calcium channels control most functions that are triggered by action potentials in excitable cells, or something similar.

Lines 137-142. More description of the conceptual basis for these methods for non-MD expects would be good here. It will be important for general readers to understand why they should accept these models as good representations of corresponding functional states.

Line 142. Figure 6 is cited out of order here. Perhaps it should be a Supplementary Figure.

Lines 139-142. Trajectories of 100 ns raise the concern that they may not accurately represent events that actually take place on the 100 usec to msec time scales in physiology. The rationale for use of Umbrella sampling to bridge these time scales should be made more clear.

Lines 120-121. Are there corresponding differences among the VSD's in interactions of their gating charges with the intracellular negative cluster, as indicated here for the extracellular interactions? Do you expect that these intracellular interactions also contribute to control of kinetics and voltage dependence?

Lines 158-159 and Figure 2. How were activated and resting states designated? Is there functional evidence to back up the designations of Resting State 2 and Resting State 3 as resting vs. active? Can one have confidence that states generated by thermal perturbation in MD analysis will reflect the same states that result from voltage stimulation?

Lines 163-164. A "shifting stretch" of 3-10 helix. Are all of the gating charges included in this 3-10 helical segment or only the central ones? Does S4 remain in 3-10 conformation after passing through the HCS? Are these conformations similar in VSD IV?

In addition to the S4 segment and its gating charge position and interactions, the previous resting state structure of the prokaryotic sodium channel revealed new conformations of the intracellular end of the S4 segment and the S4-S5 linker. Are these conformations observed in these molecular models of the resting state?

Figure 3. In most of the manuscript, the order of states begins with Activated and proceeds to Resting, presumably because that is where the MD runs start. However, it feels backwards to me. This is especially true in Figure 3 where changing the order to Resting>Activated from left to right will give a much more intuitive reading of the changes of ion pair partners in the color code.

Lines 253-254. Here the paired mutations cause a 50-fold increase in the rate of activation in the MSM model, whereas the actual mutation only causes a 4-fold the experimental results. What accounts for this large discrepancy?

Figure 5. There is a larger dispersion of WT data for activation rate in Figure 4D and in Figure 5O and Q. Why are these data more variable?

Lines 400-402. Please justify the forces used: 80 and 50 kcal/mol*rad2

[Editors' note: further revisions were suggested prior to acceptance, as described below.]

Thank you for submitting your article "Structural determinants of voltage-gating properties in calcium channels" for consideration by *eLife*. Your article has been reviewed by 3 peer reviewers, one of whom is a member of our Board of Reviewing Editors, and the evaluation has been overseen by Richard Aldrich as the Senior Editor. The following individuals involved in review of your submission have agreed to reveal their identity: Vladimir Yarov-Yarovoy (Reviewer #2); William Catterall (Reviewer #3).

Essential Revisions:

Your manuscript has been seen by all of the original reviewers, who agree that many of the issues have been resolved. However, there remains some concern about the possible dependence of sampling for the MSM model on the original umbrella sampling, and that this may have been too easily dismissed by the authors. There is still need for clarification and justification of methods, and improved illustration of tests. However, although umbrella sampling simulations may not have reached equilibrium and may still influence the subsequent dynamics to some degree, this does not necessarily preclude a valid MSM, especially for a process that is fairly well constrained (such as in this case where there are strong interactions between the S4 and other VSD residues during S4 translocation), there is sufficient MD sampling achieved after the umbrella sampling, the sampling is validated with Chapman-Kolmogorov tests, and the data is supported by experimental measurements. Thus, although ideally data would be provided that proves the final results do not depend on the initially pulling procedure (redoing with a different starting procedure/path), we do not insist on this here. We do, however, ask that: 1. the possibility that results may depend on those choices not be dismissed; 2. remaining questions and comments from the reviewers regarding clarity of methods and results be addressed in the revised manuscript; and 3. please provide the requested PDB models.*Reviewer #1:*

The authors have addressed some concerns, but the major critique regarding the possible dependence of results on the initial umbrella sampling and the description of methods have not been fully resolved. The authors did not seem to understand the concern expressed about the possible dependence of results on the initial umbrella sampling path. We know that the authors analysed MD with their MSM after US, but the point is that all sampling commenced with a pulling of the S4 along a predetermined path using umbrella sampling, done so because of the long timescale of sampling S4 movements, and one cannot just expect the subsequent MD not to be biased by that initial pulling.1. The authors admit that the umbrella sampling was used to seed the subsequent MD because otherwise the long timescales of S4 movement may not be reached, but then dismiss the possibility that the subsequent MD may remain trapped in the vicinity of that umbrella sampling path? The authors further dismiss this possible dependence by saying that this is because of their inherent statistical checks, which does not prove much to me. Statistical analysis on the subsequent MD may show reproduction of that MD by the MSM, but may still be restricted to their sampling close to the initial umbrella sampling path. I note that the cartoon in Figure S10 illustrates (perhaps with actual data, it is unclear) that the sampling from MD looks very similar in pattern to the original umbrella sampling, highlighting the concern. To say that all MSM analysis of the MD after umbrella sampling was independent of the initial US because it came after the umbrella sampling is missing the point. One could instead model initial S4 movement along an extremely different pathway (e.g. one that involves helix rotation and interaction breaking/forming before or after translocation in different stages…, perhaps using more than one translocation coordinate) and then go off sampling a different statistically reliable set of sampled states, if those different initial pathways are kinetically separated.

2. The methods section still does not explain all approaches clearly. What is the mysterious "dihedral torsion restraint" on S4, why was it needed and what was its purpose? Is it because the pulling of S4 caused the helix to lose secondary structure? What atom selections did it involve? The authors merely write that it was used to "minimise local artifacts"! Details of the umbrella sampling and its reliability remain unclear. The 1D umbrella sampling was along a translocation coordinate, presumed parallel to the helix, and how do we know such a pulling was successful? I note that S4 was pulled with US away from equilibrium, at 1 Å per 20ns (see later note on constraint also). If this trajectory is meant to yield states near equilibrium to seed the subsequent MD (as opposed to a steered MD pulling which then relies on the following unbiased MD to reach equilibrium, where equiuiibration would need to be discarded prior to MSM analysis), then it is essential that convergence of the umbrella sampling itself is demonstrated (e.g. a profile of free energy along the chosen order parameter for time 0-T with T increasing, showing convergence to within a reasonable tolerance like 1 kcal/mol). If the authors were to redo the umbrella sampling, pulling in the opposite direction, I would expect significant hysteresis. If the pulling never reached equilibrium, why should we have confidence that the subsequent MD used for MSM analysis was not influenced by this?

3. Regarding the tests in Figure S4. This figure is poorly described in its caption. The purpose, approach and what are each of the panels for the Chapman-Kolmogorov tests should be properly described. Moreover, the figure has tiny information at low resolution such that it is hard to see what is being shown in each of the small panels.

4. The mix up of energy units was unfortunate. I still am not confident all units are right. Free energies are now in kJ/mol but I note constraints are still specified in kcal/mol. The pulling of S4 by residues R1, R2 and R3 uses a constraint of 80 kcal/mol/A^2. If correct (not kJ/mol), this is an extremely tight constraint. Assuming equilibrium and applying equipartition theorem, this restraint would correspond to a standard deviation of less than 0.1 Å, such that windows spaced at 1 Å cannot overlap at all (probability of overlap at 5-10 standard deviations being vanishingly small). I suggest overlap can only be obtained as a result of the non-equilibrium pulling periods of the trajectory. The umbrella sampling was therefore used only as a non-equilibrium pull and not to establish an equilibrium distribution prior to unbiased MD.

Reviewer #2:

Authors addressed all of my comments, except providing PDB coordinates of Ca_V_1.1 channel models. I recommend to accept this manuscript for publication in *eLife* once the authors will provide PDB coordinates of Rosetta models of VSDI and VSDIV only – no need to provide the full Ca_V_1.1 channel models.

Reviewer #3:

The authors have provided comprehensive new information on their methods and have further validated their conclusions with statistical summaries. They have responded effectively to my comments, and I now recommend acceptance for publication.

---

## [Author Response]

Essential revisions:The reviewers are concerned about the heavy reliance on simulations that are not well described and lack measures of reliability. Better description of the conceptual basis of the methods for non-MD expects and much more precise definition of the methods for MD-experts are needed so that the readers can accept that the models are reasonable representations of the gating motions studied.The reviewers criticise the lack of statistical reliability tests, with absence of measures of error, convergence and reproducibility, each of which are required for publication in eLife. This is particularly important as trajectories of the order of 100 ns need to represent events that actually take place on 100 microsecond or longer time scales in physiology, such that simulations could be merely anecdotal, incomplete, dependent on starting models or dependent on the choices made in enhanced sampling techniques. Currently, the description leaves the reader to guess what has been done, unsure of the impacts those decisions might have on the findings, as well as the strengths and weaknesses of the approaches. Thus, as well as detailed responses to the comments of the reviewers, it is essential that the simulation methods be better described and justified, with results supplemented by rigorous proofs of sampling (e.g. interconversion, back and forth, between identified macrostates), convergence (e.g. MSM results changing to within a small tolerance) and reproducibility (e.g. not being dependent on the selected initial Umbrella Sampling), along with quantitative comparisons to experiment.

We thank the editors and reviewers for their exceptionally thorough review of our article. We are equally grateful for your overall encouraging assessment of our study, as for numerous valuable questions and recommendation. In compliance with the above-listed essential revisions, we re-wrote the description of our modeling approach in the Results section, to better describe the conceptual basis of our modeling approach, comprehensible also for the non-MD experts. Moreover, in the Materials and methods we provide more details and explanations so that the experts will be able to understand and possibly reproduce the study. In the Supplements we now also provide additional figures and tables showing the confidence intervals and the reliability tests for the Markov-state models.

Specifically:

Sampling and convergence: Implied time scale plots and Chapman-Kolmogorov-tests demonstrate the validity of our MSM. This and the observed connectivity imply rigorous sampling.

Reproducibility/ dependence on Umbrella sampling: We clarified that our MD simulations and the parameters deduced from it do not depend on the initial Umbrella sampling.

Quantitative comparison of modeled and experimental time constants are discussed in the text.

Importantly, the reviewer´s critique revealed a labeling error of the 1D free energy barrier estimation (kcal/mol instead of kJ/mol), which explains the inconsistencies of our data noted by reviewer #1. Correcting this error eliminated these inconsistencies and thus resolved several of the points of criticism raised by that reviewer. We apologize for this mistake and for the misunderstandings it caused when reviewing our paper.

Finally, in sum the reviewers provided a sizable list of comments and suggestions, which for the most part were implemented in the revised manuscript. For details, please see the point-to-point reply below.

Reviewer #1:This manuscript represents a study of the voltage sensor movements in homology models of Ca_V_1.1 to explain its slow kinetics and right shifted V dependence in terms of the roles of distinct structures and ion pair interactions in VSDs I and IV, backed by electrophysiology. In particular, the results suggest interactions specific to subunit chemistry can drastically affect activation. This is backed by mutagenesis and electrophysiology seeing special roles played by residues, such as E87 affecting V dependence while E90 affects rate, representing important information on how subunits of a Ca_V_ control voltage gating. The study is well written and presents a unified story to explain kinetics and V dependence. The models are used in MD simulations to examine thermodynamics and kinetics of the voltage-sensing transitions, but they are not well described and reliability tests appear non-existent, as discussed more below. A more involved examination of sampling, reproducibility and errors is needed to trust those results. Without this it is hard to judge if they are even qualitatively consistent with the electrophysiology to provide insight into voltage sensing.

We appreciate the reviewer´s overall positive assessment and we agree that the original manuscript was in want of more detailed description of the modeling methods and information about sampling, reproducibility and errors of the results. We addressed this point making substantial revisions, which are described in detail below.

Although given a fairly wordy description in the methods, the MD approaches are not well described and I am left unsure of what was actually done. Umbrella sampling was first used to "overcome high barriers". But details of the Umbrella Sampling are not described. I assume this was 1D sampling, but what is the reaction coordinate? On page 18 the authors talk about "pulling S4 down" with K = 80 kcal/mol*rad*2. What does that even mean having an angular restraint? Likewise, the authors mention some torsion restraint on S4, but it is not clear what it was actually acting on and its purpose. Was this the restraint for each window? How many windows and how was the coordinate defined? Was each of many windows run for 100ns or was that just subsequent sampling in a few macrostates? Did the windows overlap well and was the preliminary free energy profile from this Umbrella Sampling (not provided) converged? As this Umbrella Sampling set the stage for all of the MD results, how it was carried out is critical. Importantly, as the authors admit the barriers are high and timescales long for S4 movements, if not captured adequately by the chosen reaction coordinate may lead to far-from-equilibrium starting points for subsequent simulations, tICA analysis and MSM. (n.b. The outward sliding motion of S4 is important for analysis and interpretation, and while this may be the case, until we know how the starting structures were made with US we cannot tell if it is an artifact of the starting trajectory or not.)The authors then took clusters and simulated different "activation states" for 100ns each. Does that mean activated states, or activation states between activated and resting states? How were these states actually identified? We would need to see those defined states from the Umbrella simulations and how they were chosen. Are final results depending on such decisions?

To address this criticism, we extended the methods section so that it becomes clear what exactly has been done. Umbrella sampling was used exclusively to generate a potential pathway of the downward movement of the S4. This is necessary to compensate for the force of the membrane potential gradients lacking in our model, because it is based on the starting structure of Ca_V_1.1 solved in the depolarized state. Experimental details including the reaction coordinates and the justification of torsional restraints are now described in the Materials and methods (pages 2021).

Importantly, all other results, including the free energy surfaces, do not rely on the Umbrella sampling, as we used this pathway only to generate starting structures for classical molecular dynamics simulations in an extended region of the phase space (i.e., along the expected trajectory of S4 during the downward movement). The states are derived from the Markov State Model (i.e., representing the free energy wells) and are reliable within its statistical significance (see new Table S1). Therefore, the presented/analyzed structures are independent of the initial Umbrella sampling. This important point has now been made clear in the description of the approach in the Methods section (page 20 and Figure S10).

The methods description following this stage is vague. The authors say on page 19 that they "discretize the obtained conformational space into so-called microstates, grouping together conformations of the system that can exchange rapidly (e.g., k-means clustering)." Most eLife readers will not understand this, and it is vague in that it does not specify the actual method, but some example method "k-means clustering" without explaining what it means. They then say "To create a more understandable model, a kinetic clustering of a relevant set of microstates to so-called macrostates, can be performed, which are larger aggregates that correspond to the free energy wells (e.g., PCCA+ clustering)." Again stating an example method called PCCA+ clustering. Was this what was used, and what does it mean? Then, MSM is used to extract thermodynamics and kinetics. Details of this analysis, its reliability and reproducibility are completely lacking.

We are grateful to the reviewer for these comments and questions and addressed them as follows: We extended the Methods section, adding explanations for all the applied algorithms (k-means clustering, PCCA clustering, Markov-state models). We also included the Chapman-Kolmogorov plots and implied timescale plots (Figure S4), which are algorithms to estimate the reliability of the Markov-state models.

In Figure 2 and later figures, the authors choose to plot sampling and identify states in the dominant two tICA vectors. What do these vectors actually mean physically? How do we know only 2 such order parameters characterise the slowest coordinates in the activation transition? Subsequent panels in Figure 2 suggest an activation/deactivation reaction coordinate that is visualised as a sliding S4, presumably due to changing structures in the macrostates. How does concerted movement between minima in tICA1,2 translate to this S4 sliding movement? Those two order parameters would need visualisation to become meaningful to the reader.

In the revised manuscript we also included a detailed explanation of tIC vectors. The first two vectors correspond to the kinetically slowest directions in cartesian coordinate space and thus capture the majority of the slow movements (typically 70% of the movements slower than the lag time). The S4 sliding movement is primarily represented by tIC1, whereas tIC2 corresponds more to reorganization processes along this movement. A schematic structure showing the two motions represented by the tIC1 and tIC2 coordinates is now included in Figure S10.

Throughout the paper the data is presented without any errors. e.g. What are the error bars in transition mean first passage times in Figure 2? This is particularly important for barriers and the exponentially sensitive rates. Figure 2d suggests activation barriers of 60 kcal/mol. Is that even possible? If one were to estimate a rate for hopping over such barriers, it would not be microseconds (more likely millennia)! How reliable are those barriers (and estimated rates)? Analysis to calculate errors and prove evidence for reproducibility of the results is missing.

As requested, we now show the confidence intervals for the calculated mean first passage times in the Supporting Methods to show the statistical relevance of our results (Table S1). We calculated the barriers in Figure 2d based on the obtained mean first passage times, to visualize the results in a different way.

For estimating the energy barriers we used the transition state theory, the equation of which is now included also in the Supplementary Material. Unfortunately, we had specified the wrong units for the estimated 1D free energy barriers. The results were illustrated in kJ/mol, but had been mislabeled as kcal/mol in the original manuscript. We apologize for this mistake. This error has now been corrected in all plots (Figures 2, 4, and 5) and throughout the text.

Regarding these barriers, I have concerns about sampling of interconversion of macrostates to capture the MFPT. e.g. Looking at Figure 2a, it seems that there is no sampling between them in the chosen TICA1-2 space, and this raises doubt about the free energies and rates. Regarding apparent lack of overlap between sampling in macrostates (most apparent in Figure 2A), what is the origin of this? This may come down to the Umbrella Sampling used to get starting points for subsequent 100ns simulations for MSM analysis. Some reaction coordinate (unspecified) was presumed for Umbrella Sampling and pulling along that direction may have yielded patches of sampling such that the minima found in the MSM are artifacts of the starting Umbrella trajectory (sequence of windows). Would the same results emerge from a different initial starting method?

Indeed, in Figure 2a there is sampling between the macrostates, which is a prerequisite for the construction of the MSM. The sampling is validated by the inherent statistical checks in the course of MSM building. Thus, consistent results would emerge from different initial starting methods.

In Figure 2 panel A, the scale goes to 10kT whereas the predicted barriers are much higher (60 kcal/mol). These are given different units, and 60 kcal/mol is of order 100kT, being on a different order to the scale of the free energy maps. n.b. On page 9 the authors now refer to barriers of 50 kJ/mol (a third different unit of energy and again different magnitude). This of order 10-15 kcal/mol and appears inconsistent with the above data. Units again on page 10 in kJ/mol appear wrong. I do not trust the predictions based on the data shown. We would need to be convinced of good sampling of transitions both back and forth between macrostates, such that the MFPT estimates have high accuracy, with propagated errors to rates estimated. Regarding rate estimates: How is it that the predicted barriers between successive macrostates change by 10 kcal/mol (e.g. sequence of barriers in panel D), but rates between macrostates (e.g. in panel B) remain of the same order of magnitude (1ms)? This makes no sense to me. Perhaps the barriers in panel D and other similar graphs are not based on any calculated data?

This inconsistency resulted from the unit error stated above. We corrected the units of the tIC plots to kJ/mol. Consequently, it is now apparent that the free energies presented in the tIC surfaces are in the same order of magnitude.

The authors write that "although the absolute times derived from MSM may not correspond to the actual transition times of the VSD upon physiological activation and deactivation, relative differences between transition times provide meaningful information when compared between different VSDs or functionally different mutants". Sources of error in absolute times (and comparison to known experimental times) is required. Regarding reliability of relative rates, how do we know to believe the above statement? Besides model and force field dependence, proof of error and reproducibility of what was sampled is needed.

We totally agree with the reviewer and now provide the confidence intervals for all transition times and also the reliability checks for the underlying Markov-state models (Table S1, Figure S4).

On page 11 the authors state that "additionally formed ion-pairs in the restingstates of VSD I are paralleled by a remarkable increase in the energy barriers and the state transition times, suggesting that the number and strength of interactions between the gating charges and the ENC transiently formed in the resting states determine the slow activation kinetics of VSD I." It is suggested here that those changed interactions change the barriers, as indicated in Fig2D-H by 20 kcal/mol. Increased barrier like that means a prefactor (assuming similar order of friction) of exp(-20kcal/mol / k_B T) of the order of 10^-14. Yet the rates change between just micro and millisecond. Again, perhaps the representation of the barriers in the figures are not actual computed barriers? Related, on page 13 the authors write that their result "substantiates the reliability and predictive value of the kinetic analysis of our MD simulation". A much more quantitative comparison than this is needed to demonstrate reliability.

Again, this inconsistency arose from our mislabeling of the units. This has been corrected and the results are now in line with the timescales and the free energy plots. Obviously, as pointed out by the reviewer, the largest errors have to be expected for kinetics due to the exponential dependency on barriers.

Reviewer #2:Manuscript by Monica Fernández-Quintero et al. describes study of molecular mechanism of voltage-gated calcium channel gating using experimental and computational modeling approaches. Authors identified specific ion-pair interactions stabilizing either activated or resting states of voltage sensing domains I and IV in eukaryotic voltage-gated calcium channel. Overall, the results are promising and provide evidence towards evolutionary functional conservation of the sliding helix hypothesis. This study provides key structural insights into voltage-dependence and kinetics of current activation of eukaryotic voltage-gated calcium channels.1. Discuss potential role of other interactions involving hydrophobic and polar residues in stabilizing specific VSD states besides the ion-pair interactions identified in this study. Also note that residue size and hydrophobicity may also contribute to differences in VSD kinetics – see Jérôme J. Lacroix et al. (2014) Moving gating charges through the gating pore in a K_V_ channel voltage sensor. Proc Natl Acad Sci U S A, 111(19): E1950-9. doi: 10.1073/pnas.1406161111.

We agree with the reviewer that the movement of S4 through the VSD involves many more interactions than the ion-pair interactions of the ENC studied here. This fact now is explicitly acknowledged in our description of the ionic interactions (Results, lines 180-183)

2. A large part of the study is computational modeling to generated models of VSDs in different resting states and run MD simulations to compute transition kinetics. However, to prove the rigor of the approach authors need to describe essential details and related results in supplemental figures and/or text. Specifically:a. What was the reaction coordinate in the Umbrella sampling?

We extended the methods section, describing the applied protocol in detail (pages 20-21)

b. How many windows used and how well they overlapped?c. How did the authors decide if the Umbrella sampling has generated enough data for the following steps?

The Umbrella sampling was used exclusively to generate a potential pathway of the downward movement of the S4. This pathway provided the starting structures for classical MD simulations, but did not define states. Consequently, all subsequent steps and analyses, like the free energy surfaces, are independent of the data generated by Umbrella sampling. This important fact has now been clarified at several places in the article. (see pages 6-7, 20-21, and Figure S10).

d. The author mentioned for clustering the Umbrella sampling trajectory "… resulting in a large number of clusters". How many clusters exactly, given all of these clusters were subsequently simulated for 100 ns?

The number of clusters was between 46 and 56 resulting in an aggregated simulation time of about 5 µs for all variants.

e. How was the time lag of 10 ns chosen for the tlCA analysis in the MSM modeling?f. The authors should show an implied timescales (ITS) plot as a function of different lag times. Was it convergent?g. The authors should also show the results of the Chapman-Kolmogorov test to ensure the dynamics observed is Markovian so that the results can be meaningful.

In the revised manuscript we provide a more detailed description of the used methods and in the Supplements we provide the implied timescale plots and the ChapmanKolmogorov test, as requested.

3. Discuss accuracy of absolute activation and deactivation times estimated by MSM – how far away they are from the actual transition times observed experimentally?

To our knowledge activation and deactivation kinetics of individual VSD of Ca_V_1.1 are not known and probably impossible to measure directly in skeletal muscle cells. Thus, any further discussion of the differences and/or similarities of the transition times derived from MD/MSM would be highly speculative. Therefore, we suffice it to say that, “Because the values calculated in our model are obtained in the absence of the force provided by changes in the electric field, the absolute times derived from MSM may not correspond to the actual transition times of the VSD upon physiological activation and deactivation”. (line 187)

4. An assumption in the computational modeling part of this study is that transitions between VSD states can happen at equilibrium. However, this rarely happens in reality. The activation or deactivation can be considered as a non-equilibrium process by which the electric field exerts force on gating charges to trigger conformational changes in VSDs. Authors should discuss strength and weaknesses of chosen computational modeling methods.

This important limitation has now been explicitly stated in the sentence quoted at the end of the previous paragraph. (line 187)

5. Note that while the experimental testing is in agreement with observations form computations models, it is not a direct validation of accuracy of the computational prediction. Given the largely unchanged conformation of the rest of the channel except S4, the computational modeling at its best can only infer S4 gating charges movement in individual VSD in a specific physiological state of the channel. The experiments, on the other hand, only showed macroscopic view of the channel activation, in which the contribution of the gating charges movement, VSDs cooperation, and pore coupling is complex. The authors need to discuss these limitations.

We agree, this is an obvious but important issue. Therefore, we have now added a brief discussion of this limitation and its consequences to the interpretation of our MD/MSM results vis-à-vis channel gating data. (lines 275-283)

6. The authors studied activation and deactivation kinetics of the gating charges movement using the computational modeling approaches. Discuss why only activation kinetics have been studied experimentally.

To address this issue based on hard data, we conducted additional experiments and analyzed the deactivation kinetics of WT VSD I and the E87A and E90A mutants (new Supplementary Figure S7). As expected, in all cases, the deactivation kinetics were fast and in the range of the activation kinetics observed in the fast constructs (E90A, E87A/E90A). Fast deactivation is expected even in the slowly activating constructs, because closure of the gate is determined by the first (fastest) VSD moving from the activated state into resting state 3, rather than by the slow VSD I, which is rate limiting only during activation. This is now discussed at the end of the Results section (lines 312-325).

7. Can authors provide computational ion-pair energy calculations to support VSD state-stabilization hypotheses?

We performed LIE calculations (linear interaction energy calculations implemented in cpptraj) to calculate the energies of electrostatic interactions of the S4 helix with the other parts of the voltage sensors for the macrostate representatives of the two VSDs and the mutants. We provided a table in the Supporting information (Table S3).

8. Provide in the Supplementary Information example scripts and command lines used for Rosetta homology modeling, MD simulations, and MSM calculations. Such scripts should contain enough information such that someone else could independently run and validate author's data.

We provide a more detailed description of our approach in the Methods section as well as the input command lines for the loop modelling and the MD simulations (see Supplementary Materials, last 2 pages). For the Markov-state models we used the protocol presented in the PYEMMA tutorials (http://www.emma-project.org/latest/tutorial.html#jupyter-notebooktutorials) including all validation tests.

9. Provide confidence intervals for experimental data.

All our patch-clamp data already showed descriptive statistics (scatter plots with mean and SEM). For the MSM and transition kinetics we now also provided reliability tests and confidence intervals in the Supplements (Table S1 and Figure S4).

10. Provide PDB coordinates of Rosetta Ca_V_1.1 models as text files in the Supplementary Information.

We chose not to provide the PDB coordinates of the complete Ca_V_1.1 in this paper, because it also contains critical information on parts of the channel, which are not relevant here but important for independent ongoing work.

Reviewer #3:Fernandez-Quintero et al. show how changes in the negatively charged amino acid residues in the voltage sensor of skeletal muscle Ca_V_1.1 calcium channels can control their function. These calcium channels serve two physiological functions with different kinetics and voltage dependence: excitation-contraction coupling triggered by conformational coupling with the ryanodine receptor in the sarcoplasmic reticulum and calcium entry triggered by opening of their pore. Voltage-driven movement of the S4 segments in their voltage sensors initiate both functions. Molecular models are developed here for the function of VSD I and VSD IV describing the transmembrane movements of their S4 segments from resting to activated states. In each case, the molecular models fit a sliding helix mechanism of VSD function in which the S4 segments exchange ion pair partners between the extracellular negative cluster and the intracellular negative cluster of counterions in the voltage sensor. The kinetic barriers for movement of the S4 segment in VSD IV were much lower than VSD I, suggesting that VSD I controls the slow activation of the calcium conductance of Ca_V_1.1 channels. Introducing mutations in two negatively charged amino acid residues in the extracellular negative cluster of VSD I (E87A and E90A) showed that they differentially regulate channel function. E87A positively shifted the voltage dependence of activation, whereas E90A gave a smaller positive shift in activation and slowed the rate of activation by four-fold.Understanding how molecular specializations in VSDs control ion channel function is an important goal of molecular physiology. Because of its unique functional properties, the Ca_V_1.1 channel provides an interesting experimental model. The results presented here are an important advance, as they provide a detailed new molecular model of VSD function and they show how a conserved molecular feature of VSDs, the negatively charged amino acid residues in the extracellular negative cluster, control the voltage dependence and rate of activation of VSD I.Specific Comments:Line 33. It is an overstatement to say calcium channels control most functions in excitable cells. Many hormone receptors and other regulators act independently of them. It seems more accurate to say that calcium channels control most functions that are triggered by action potentials in excitable cells, or something similar.

We replaced “most functions” with “key functions” which certainly can be maintained, considering the examples given (synaptic transmission, contraction in heart and skeletal muscle).

Lines 137-142. More description of the conceptual basis for these methods for non-MD expects would be good here. It will be important for general readers to understand why they should accept these models as good representations of corresponding functional states.

In order to make the modeling approach more accessible to non-MD experts, we amended the description of our approach in the Results section in as simple terms as possible (lines 140148). Much additional detail and the conceptual basis of the methods have now been provided in the Material and Methods and in the Supplements (lines 445-483). Furthermore, in response to the comments of reviewer #2, we included brief discussions of some of the limitations of this approach, when comparing it with channel gating data.

Line 142. Figure 6 is cited out of order here. Perhaps it should be a Supplementary Figure.

As suggested, we moved former Figure 6 to the supplements Figure S10 and adjusted the citation accordingly.

Lines 139-142. Trajectories of 100 ns raise the concern that they may not accurately represent events that actually take place on the 100 usec to msec time scales in physiology. The rationale for use of Umbrella sampling to bridge these time scales should be made more clear.

In the revised manuscript we provide a discussion of the different activation time constants observed in the model and experimentally (lines 275-283). In the Methods section we also extended the rational for using Umbrella sampling (see also Figure S10).

Lines 120-121. Are there corresponding differences among the VSD's in interactions of their gating charges with the intracellular negative cluster, as indicated here for the extracellular interactions? Do you expect that these intracellular interactions also contribute to control of kinetics and voltage dependence?

Strictly speaking, we do not know, because we have not examined this question directly.

However, the ion-pair interactions in the INC are highly conserved across VSDs of all Ca_V_, Na_V_, and K_V_ channels. Moreover, our model of the activated state does not indicate similar differences. Therefore, we consider the interactions in the INC (i.e., in the charge-transfer center) as the common (essential) mechanism of voltage-sensing, whereas our present results suggest that the ion-pair interactions in the ENC, which differ even between the VSDs of one pseudo-heterotetrameric channel, determine the specific gating properties.

Lines 158-159 and Figure 2. How were activated and resting states designated? Is there functional evidence to back up the designations of Resting State 2 and Resting State 3 as resting vs. active? Can one have confidence that states generated by thermal perturbation in MD analysis will reflect the same states that result from voltage stimulation?

The states were designated by MSM and represent the structures at free energy wells. The sampling is validated by the inherent statistical checks in the course of MSM building. Thus, assuming statistical significance, consistent results would emerge from different initial starting methods, such as voltage simulations. However, this clearly would be beyond the scope of the present study. The quality of MSM can now be assessed in the provided implied time scale plots and the Chapman-Kolmogorov plots (Figure S4).

As to the designation and credibility of the resting states, it is reassuring that the states resulting from our simulation correspond with the conceptual models based on the sequential movement of S4 through the charge-transfer center, as well as with the recently solved resting state structures of Na_V_Ab and a Na_V_Ab/Na_V_1.7 VSD II chimera. Therefore, the term

“resting states” seems a very reasonable assumption.

Lines 163-164. A "shifting stretch" of 3-10 helix. Are all of the gating charges included in this 3-10 helical segment or only the central ones? Does S4 remain in 3-10 conformation after passing through the HCS? Are these conformations similar in VSD IV?

We added a figure in the Supporting Information (Figure S3) comparing the 3_10_ helix content between the two VSDs and showing the respective gating charges.

In addition to the S4 segment and its gating charge position and interactions, the previous resting state structure of the prokaryotic sodium channel revealed new conformations of the intracellular end of the S4 segment and the S4-S5 linker. Are these conformations observed in these molecular models of the resting state?

To accomplish computing of the MD simulations, we analyzed the isolated voltage-sensing domains without the S4-S5 linker. Analyzing the mechanical transduction of the VSD motion to opening and closing the channel gate will require MD simulations of the complete channel. This is an important goal that shall be tackled in a next step.

Figure 3. In most of the manuscript, the order of states begins with Activated and proceeds to Resting, presumably because that is where the MD runs start. However, it feels backwards to me. This is especially true in Figure 3 where changing the order to Resting>Activated from left to right will give a much more intuitive reading of the changes of ion pair partners in the color code.

We agree that from a physiological standpoint depicting the states from resting to activating would make more sense. However, as the reviewer stated, the available high-resolution structures of VSD are in the (activated) up-state and this is where our simulations start. Accordingly, also the expected accuracy of the modeled states will be highest in the resting state 3. Therefore, we prefer to present the states in this order and, for consistency reasons, stick to this order in Figure 3 as well.

Lines 253-254. Here the paired mutations cause a 50-fold increase in the rate of activation in the MSM model, whereas the actual mutation only causes a 4-fold the experimental results. What accounts for this large discrepancy?

On the one hand, we only considered the individual voltage-sensing domains and on the other hand any errors in the values of kinetics would be potentiated due to their exponential dependency on the energy barriers. We also provided errors on the transition timescales. We added a paragraph discussing that aspect.

At least in part, these differences in magnitude result from the fact that MD/MSM determines the kinetics of a single VSD, whereas the current properties reflect the concerted action of the entire channel, the gating of which is differentially determined by four VSDs and the transduction of their action to the channel gate. This has now been clarified in the Results (line 275-283)

Figure 5. There is a larger dispersion of WT data for activation rate in Figure 4D and in Figure 5O and Q. Why are these data more variable?

Cell-to-cell variations arise from many factors including differences in the size and differentiation of the reconstituted myotubes and from the expression efficiency. This is normal. Therefore, we perform our recordings with matched controls from the same cell passages and transfections.

Lines 400-402. Please justify the forces used: 80 and 50 kcal/mol*rad2

80 and 50 kcal/mol*rad2 was determined by trial and error to allow efficient sampling while minimally distorting the system. In the revised manuscript (Methods section) we give a justification of the applied forces.

[Editors' note: further revisions were suggested prior to acceptance, as described below.]

Essential Revisions:Your manuscript has been seen by all of the original reviewers, who agree that many of the issues have been resolved. However, there remains some concern about the possible dependence of sampling for the MSM model on the original umbrella sampling, and that this may have been too easily dismissed by the authors. There is still need for clarification and justification of methods, and improved illustration of tests. However, although umbrella sampling simulations may not have reached equilibrium and may still influence the subsequent dynamics to some degree, this does not necessarily preclude a valid MSM, especially for a process that is fairly well constrained (such as in this case where there are strong interactions between the S4 and other VSD residues during S4 translocation), there is sufficient MD sampling achieved after the umbrella sampling, the sampling is validated with Chapman-Kolmogorov tests, and the data is supported by experimental measurements. Thus, although ideally data would be provided that proves the final results do not depend on the initially pulling procedure (redoing with a different starting procedure/path), we do not insist on this here. We do, however, ask that: 1. the possibility that results may depend on those choices not be dismissed; 2. remaining questions and comments from the reviewers regarding clarity of methods and results be addressed in the revised manuscript; and 3. please provide the requested PDB models.

In the revised manuscript we addressed all three points requested above.

Reviewer #1:The authors have addressed some concerns, but the major critique regarding the possible dependence of results on the initial umbrella sampling and the description of methods have not been fully resolved. The authors did not seem to understand the concern expressed about the possible dependence of results on the initial umbrella sampling path. We know that the authors analysed MD with their MSM after US, but the point is that all sampling commenced with a pulling of the S4 along a predetermined path using umbrella sampling, done so because of the long timescale of sampling S4 movements, and one cannot just expect the subsequent MD not to be biased by that initial pulling.

We do not claim that this approach is unbiased. However, as nicely summarized in the “Essential Revisions” paragraph above, this does not preclude a valid MD simulation and MSM, because the experimentally conditions are solidly based on published structure-function data of VSD action, validated by appropriate tests as well as by the mutagenesis experiments shown in the article.

1. The authors admit that the umbrella sampling was used to seed the subsequent MD because otherwise the long timescales of S4 movement may not be reached, but then dismiss the possibility that the subsequent MD may remain trapped in the vicinity of that umbrella sampling path? The authors further dismiss this possible dependence by saying that this is because of their inherent statistical checks, which does not prove much to me. Statistical analysis on the subsequent MD may show reproduction of that MD by the MSM, but may still be restricted to their sampling close to the initial umbrella sampling path. I note that the cartoon in Figure S10 illustrates (perhaps with actual data, it is unclear) that the sampling from MD looks very similar in pattern to the original umbrella sampling, highlighting the concern. To say that all MSM analysis of the MD after umbrella sampling was independent of the initial US because it came after the umbrella sampling is missing the point. One could instead model initial S4 movement along an extremely different pathway (e.g. one that involves helix rotation and interaction breaking/forming before or after translocation in different stages…, perhaps using more than one translocation coordinate) and then go off sampling a different statistically reliable set of sampled states, if those different initial pathways are kinetically separated.

We explain (there is nothing to “admit”) that to overcome the timescale limitation of MD simulations in the absence of the driving membrane potential, some kind of enhanced sampling technique (i.e. Umbrella sampling) is necessary to restrict the MD simulation of the VSD to the general vicinity determined by the probable path of the S4 helix relative to the other helices. It was not our intension to “dismiss” the possibility that different starting points could result in models with extremely different pathways. We searched the manuscript carefully for statements that might suggest that. However, nowhere in our manuscript do we claim that all MD/MSM analysis after Umbrella sampling was independent of the initial US. We only write that no states were pre-defined based on US. To make this point absolutely clear, we further amended our explanations in the Methods section accordingly (lines 454 – 465).

We also clarify that the schematic figures in Figure S2 (earlier Figure S10) are exemplary figures and not data from the present modeling. (line 762)

2. The methods section still does not explain all approaches clearly. What is the mysterious "dihedral torsion restraint" on S4, why was it needed and what was its purpose? Is it because the pulling of S4 caused the helix to lose secondary structure? What atom selections did it involve? The authors merely write that it was used to "minimise local artifacts"! Details of the umbrella sampling and its reliability remain unclear. The 1D umbrella sampling was along a translocation coordinate, presumed parallel to the helix, and how do we know such a pulling was successful? I note that S4 was pulled with US away from equilibrium, at 1 Å per 20ns (see later note on constraint also). If this trajectory is meant to yield states near equilibrium to seed the subsequent MD (as opposed to a steered MD pulling which then relies on the following unbiased MD to reach equilibrium, where equiuiibration would need to be discarded prior to MSM analysis), then it is essential that convergence of the umbrella sampling itself is demonstrated (e.g. a profile of free energy along the chosen order parameter for time 0-T with T increasing, showing convergence to within a reasonable tolerance like 1 kcal/mol). If the authors were to redo the umbrella sampling, pulling in the opposite direction, I would expect significant hysteresis. If the pulling never reached equilibrium, why should we have confidence that the subsequent MD used for MSM analysis was not influenced by this?

We replace the term “dihedral torsion restraint” by a more detailed explanation of the applied backbone restraint, i.e., ϕ torsion angle, also describing the magnitude and an explanation of why this was important.

Concerning convergence, we now explicitly state that US does not result in equilibrium distributions because of insufficient overlap between the individual sampling windows, as we did not aim at a converged sampling from the US procedure. (lines 449-459)

3. Regarding the tests in Figure S4. This figure is poorly described in its caption. The purpose, approach and what are each of the panels for the Chapman-Kolmogorov tests should be properly described. Moreover, the figure has tiny information at low resolution such that it is hard to see what is being shown in each of the small panels.

We amended the figure legend to describe the panels and we increased the font size in the figure. (lines 771-775)

4. The mix up of energy units was unfortunate. I still am not confident all units are right. Free energies are now in kJ/mol but I note constraints are still specified in kcal/mol. The pulling of S4 by residues R1, R2 and R3 uses a constraint of 80 kcal/mol/Å^2^. If correct (not kJ/mol), this is an extremely tight constraint. Assuming equilibrium and applying equipartition theorem, this restraint would correspond to a standard deviation of less than 0.1 Å, such that windows spaced at 1 Å cannot overlap at all (probability of overlap at 5-10 standard deviations being vanishingly small). I suggest overlap can only be obtained as a result of the non-equilibrium pulling periods of the trajectory. The umbrella sampling was therefore used only as a non-equilibrium pull and not to establish an equilibrium distribution prior to unbiased MD.

The unit and magnitude of the constraint are correct (80 kcal/mol*Å^2^). We tested the applied parameters to avoid unfolding of the S4 helix but to have enough force to induce the downward movement of the S4. As correctly stated by the reviewer and as mentioned above, there is indeed insufficient overlap between the individual sampling windows for US to result in an equilibrium distribution as we do not aim at a converged sampling from the US procedure. This has now been explicitly stated. (lines 454-459)

Reviewer #2:Authors addressed all of my comments, except providing PDB coordinates of Ca_V_1.1 channel models. I recommend to accept this manuscript for publication in eLife once the authors will provide PDB coordinates of Rosetta models of VSDI and VSDIV only – no need to provide the full Ca_V_1.1 channel models.

As requested we now uploaded also the PDBs of VSD I, VSD IVe and VSD IVa (in addition to the already published PDBs of the VSD I WT and mutant activated and resting states structures). (https://doi.org/10.5061/dryad.hhmgqnkfd).